# Active preference learning for ordering items in- and out-of-sample

**Herman Bergström**[*]
Chalmers University of Technology
and University of Gothenburg
`hermanb@chalmers.se`

**Emil Carlsson**[*†]
Sleep Cycle AB
Chalmers University of Technology
and University of Gothenburg

**Devdatt Dubhashi**
Chalmers University of Technology
and University of Gothenburg

**Fredrik D. Johansson**
Chalmers University of Technology
and University of Gothenburg

## Abstract

Learning an ordering of items based on pairwise comparisons is useful when items are difficult to rate consistently on an absolute scale, for example, when annotators have to make subjective assessments. When exhaustive comparison is infeasible, actively sampling item pairs can reduce the number of annotations necessary for learning an accurate ordering. However, many algorithms ignore shared structure between items, limiting their sample efficiency and precluding generalization to new items. It is also common to disregard how noise in comparisons varies between item pairs, despite it being informative of item similarity. In this work, we study active preference learning for ordering items with contextual attributes, both in- and out-of-sample. We give an upper bound on the expected ordering error of a logistic preference model as a function of which items have been compared. Next, we propose an active learning strategy that samples items to minimize this bound by accounting for aleatoric and epistemic uncertainty in comparisons. We evaluate the resulting algorithm, and a variant aimed at reducing model misspecification, in multiple realistic ordering tasks with comparisons made by human annotators. Our results demonstrate superior sample efficiency and generalization compared to non-contextual ranking approaches and active preference learning baselines.

## 1 Introduction

The success of supervised learning is built on annotating items at great volumes with small error. For subjective assessments, however, assigning a value from an arbitrary rating scale can be difficult and prone to inconsistencies, causing many to favor *preference feedback* from pairwise comparisons (Yannakakis and Martínez, 2015; Christiano et al., 2017; Ouyang et al., 2022; Zhu et al., 2023). Preference feedback is sufficient to learn an *ordering* of items (Fürnkranz and Hüllermeier, 2003), but for $n$ items, there are $\mathcal{O}(n^2)$ possible pairs of items to compare. A common solution is to use crowd-sourcing (Chen et al., 2013; Yang et al., 2021; Larkin et al., 2022), but many tasks require domain *expertise*, making annotations *expensive* to collect. This is the case in the field of medical imaging, where annotations require trained radiologists (Phelps et al., 2015; Jang et al., 2022; Lidén et al., 2024; Tärnåsen and Bergström, 2023). So, how can we learn the best ordering possible from a limited number of comparisons?

---

[*]Equal contribution. [†]Work was mainly performed while the author was a PhD student at Chalmers University of Technology.

38th Conference on Neural Information Processing Systems (NeurIPS 2024).

Classically, this problem is solved by active learning, sampling comparisons based on preference feedback and estimated item scores (Herbrich et al., 2006; Maystre and Grossglauser, 2017; Heckel et al., 2018). However, consider a radiologist who wants to quantify the expression of a disease in a collection of X-ray images. Purely preference-based algorithms utilize only the outcomes of comparisons but ignore the contents of the X-rays, which can reveal similarities between items and inform an ordering strategy. Moreover, the set we want to order is often larger than the set of items observed during training—we may want to rank new X-rays in relation to previous ones. This cannot be solved by learning per-item scores alone. As an alternative, active learning for classification can be used to fit a map from pairs of item contexts $x_i, x_j$ (e.g., the contents of images) to the comparison $i \succ_? j$, that can be applied to old and new items alike (Houlsby et al., 2011; Qian et al., 2015). However, as we show in Section 4, learning this map to recover a *complete ordering* is distinct from the tasks preference learning is commonly used for, and existing algorithms lack theoretical justification for this application. Moreover, formal results for related problems, such as contextual bandits or reinforcement learning (Das et al., 2024; Filippi et al., 2010; Zhu et al., 2023; Bengs et al., 2022), do not translate directly to effective active sampling criteria for ordering. There is a small body of work on learning a contextual model to recover the complete ordering (Jamieson and Nowak, 2011; Ailon, 2011) but these either assume noiseless preference feedback or that the noise is unrelated to the similarity of items, which is unrealistic for subjective assessments.

**Contributions.** We propose using a contextual logistic preference model to support efficient in-sample ordering and generalization to new items. Our analysis yields the first bound on the expected ordering error achievable given a collected set of comparisons (Section 4). This result justifies an active sampling principle that accounts for both epistemic and aleatoric uncertainty which we implement in a greedy deterministic algorithm called GURO (Section 5). We further propose a hybrid variant of the contextual preference model, compatible with GURO as well as existing sampling strategies, that overcomes model misspecification by adding per-item parameters (Section 5.1). We evaluate GURO and baseline algorithms in four diverse ordering tasks, three of which utilize comparisons performed by human annotators (Section 6). Our sampling strategy compares favorably to active preference learning baselines, and our hybrid model benefits both GURO and other sampling criteria, achieving the low variance of contextual models and the low bias of fitting per-item parameters. This results in faster convergence in-sample, better generalization to new items, and efficient continual learning when new items are added.

## 2 Ordering items with active preference learning

Our goal is to learn an ordering of items $\mathcal{I}$ according to an unobserved score $y_i \in \mathbb{R}$, defined for each item $i \in \mathcal{I}$. The ground-truth ordering of $\mathcal{I}$ is determined by a comparison function $\pi_{ij} := \mathbb{1}[y_i > y_j]$, where $\pi_{ij} = 1$ indicates that $i$ ranks higher than $j$. We assume there are no ties.

We define the *ordering error* $R_{\mathcal{I}}(h)$ of a learned comparison function $h : \mathcal{I} \times \mathcal{I} \to \{0, 1\}$ as the frequency of pairwise inversions under a uniform distribution of item pairs,

$$R_{\mathcal{I}}(h) = \frac{2}{n(n-1)} \sum_{i \neq j \in \mathcal{I}} \mathbb{1}[h(i,j) \neq \pi_{ij}], \tag{1}$$

where $n = |\mathcal{I}|$. This error is equivalent to the normalized Kendall's Tau distance (Kendall, 1948).

Hypotheses $h$ are learned from *preference feedback*—noisy pairwise comparisons $C_{ij} \in \{0, 1\}$ for items $(i, j)$ related to their score, for example, provided by human annotators. $C_{ij} = 1$ indicates that an annotator perceived that item $i$ has a higher score than $j$, i.e., that they prefer $i$ over $j$. *Our goal is to minimize the ordering error $R_{\mathcal{I}}(h)$ for a fixed budget $T \geq 1$ of adaptively chosen comparisons.*

We are interested in contextual problems, where each item $i \in \mathcal{I}$ is endowed with item-specific attributes $x_i \in \mathcal{X} \subseteq \mathbb{R}^d$. As we will see, this permits more sample-efficient ordering and learning algorithms that can order items out-of-sample, trained on comparisons of a subset of items $\mathcal{I}_D \subseteq \mathcal{I}$ and generalizing to $\mathcal{I} \setminus \mathcal{I}_D$. Ordering algorithms based *only* on preference feedback cannot solve this problem since observed comparisons are uninformative of new items.

Our *active preference learning* scenario proceeds as follows: 1) A learner is given an annotation budget $T$, a pool of items $\mathcal{I}_D \subseteq \mathcal{I}$ and item attributes $x_i$ for $i \in \mathcal{I}_D$. 2) Over rounds $t = 1, ..., T$, the learner requests a comparison of two items $i_t, j_t \in \mathcal{I}_D$ according to a sampling criterion and

receives noisy binary preference feedback $c_t \sim p(C_{ij})$, independently of previous comparisons. 3) After $T$ rounds, the learner returns a comparison function $h : \mathcal{I} \times \mathcal{I} \to \{0, 1\}$. We denote the history of accumulated observations until and including time $t$ by $D_t = ((i_1, j_1, c_1), ..., (i_t, j_t, c_t))$.

We assume that comparisons $C_{ij}$ follow a logistic model applied to the difference between item scores, $p(C_{ij} = 1) = \sigma(y_i - y_j)$, the so-called Bradley-Terry model (Bradley and Terry, 1952), which assumes linear stochastic transitivity (Oliveira et al., 2018). Throughout, $\sigma(z) = 1/(1 + e^{-z})$ and $\dot{\sigma}(z)$ its derivative at $z$. Specifically, we study the case where $y_i$ is a linear function of item attributes, $y_i = \theta_*^\top x_i$ , with $\theta_* \in \mathbb{R}^d$ the ground-truth coefficients. Thus, comparisons are determined by a logistic regression model applied to the attribute difference vector $z_{ij} := x_i - x_j$,

$$p(C_{ij} = 1) = \sigma(\theta_*^\top z_{ij}) . \tag{2}$$

We face two kinds of uncertainty when actively learning the model in (2): *epistemic* and *aleatoric*. Epistemic uncertainty, or model uncertainty, is the uncertainty about the true parameter $\theta_*$, while aleatoric uncertainty is the irreducible uncertainty about labels due to noisy annotation.

# 3    Related work

**Active Preference Learning:**    *Preference learning* (Fürnkranz and Hüllermeier, 2003; Chu and Ghahramani, 2005) is related to the problem of *learning to rank* (Burges et al., 2005; Busse et al., 2012). When using adaptively chosen comparisons it may be posed as an *active learning* or *bandit* problem (Brinker, 2004; Long et al., 2010; Silva et al., 2014; Ling et al., 2020). Non-contextual active learners, such as TrueSkill (Herbrich et al., 2006; Minka et al., 2018), Hamming-LUCB (Heckel et al., 2018), and Probe-Rank (Lou et al., 2022) produce in-sample preference orderings, but must be updated if new items are to be ranked. Contextual algorithms, such as BALD (Houlsby et al., 2011), mitigate this by exploiting item structure and Kirsch and Gal (2022) show that many recently proposed contextual active learning strategies may be unified in a framework based on Fisher information. Similarly, methods have been proposed to recover a linear preference model by adaptively sampling paired comparisons (Qian et al., 2015; Massimino and Davenport, 2021; Canal et al., 2019). Still, this setting differs from ours in that we emphasize recovering the full ordering, not perfectly estimating the parameters. While it is true that knowing the parameters is sufficient to order the list, reducing uncertainty for all parameters equally will likely be wasteful (see Section 4). Ailon (2011) offer guarantees for ordering using contextual features in the noiseless setting, while Jamieson and Nowak (2011) analyze the setting where noise is unrelated to item similarity.

**Bandits:**    Bandit algorithms with *relative* or *dueling* feedback (Yue and Joachims, 2009; Bengs et al., 2021; Yan et al., 2022) also learn from pairwise comparisons, and have been proposed both in contextual (Dudík et al., 2015) and non-contextual settings (Yue et al., 2012) to minimize regret or identify top-$k$ items. Bengs et al. (2022) proposed CoLSTIM, a contextual dueling bandit for regret minimization under linear stochastic transitivity, matching (2), and Di et al. (2023) gave variance-aware regret bounds for this setting. However, algorithms that find the top-$k$ items, such as pure exploration bandits (Fang, 2022; Jun et al., 2021), can be arbitrarily bad at learning a full ordering (see Appendix D). Related are also George and Dimitrakakis (2023) who learn Kemeny rankings in non-contextual dueling bandits, and Wu et al. (2023b) who minimize Borda regret. Zhu et al. (2023) studies the problem of estimating a preference model from offline data. Our analysis uses techniques from logistic bandits (Filippi et al., 2010; Li et al., 2017; Faury et al., 2020; Kveton et al., 2020).

**RLHF:**    Preference learning is commonly used when training large language models through reinforcement learning with human feedback (RLHF) (Christiano et al., 2017; Bai et al., 2022; Ouyang et al., 2022; Wu et al., 2023a). In this line of work, Zhu et al. (2023) provide guarantees on the sample complexity of learning a preference model from offline data. They leverage similar tools from statistical learning and bandits as we do. In contrast to their work, we provide sampling strategies for the online setting. Mehta et al. (2023) consider active learning for RLHF in a dueling bandit framework where the goal is to optimize a contextual version of the Borda regret. Concurrent work by Mukherjee et al. (2024) and Das et al. (2024) studies a similar problem, as we do here, in the RLHF setting but with the objective to identify an optimal policy in a contextual bandit with dueling feedback. In contrast to their objective, we are interested in recovering the ordering of items. Das et al. (2024) use similar bandit techniques as we do, and their selection criterion, when adapted for ordering, corresponds to our NormMin baseline (see Section 6).

## 4 Which comparisons result in a good ordering?

We give an upper bound on the ordering error $R_{\mathcal{I}}(h)$ for a hypothesis $h$ fit using the feedback from a given set of $T$ comparison queries $\mathcal{Q}_T = ((i_1, j_1), ..., (i_t, j_T))$. In other words, the bound attempts to answer the question "if we make queries $\mathcal{Q}_T$, how good can we expect our resulting model to be at ordering the items in $\mathcal{I}$?". That is, we condition on the queries themselves and reason about the uncertainty due to the stochastic feedback $c_t$. In Section 5, we use insights from the result to design an active learning algorithm.

We restrict our analysis to the logistic model in (2) and denote by $R(\theta) \equiv R_{\mathcal{I}}(h_\theta)$ the risk of the hypothesis defined by $h_\theta(i, j) = \mathbb{1}[\theta^\top z_{ij} > 0]$. Recall that $z_{ij} = x_i - x_j$ for $i, j \in \mathcal{I}$, and define $z_t \equiv z_{i_t j_t}$ as the difference between attributes for the pair of items selected at round $t = 1, ..., T$. Let $\theta_t$ be the maximum-likelihood estimate (MLE) fit to $t$ rounds of feedback, $D_t$

$$\theta_t = \arg\max_\theta \sum_{s=1}^{t} \left( c_s \log \sigma(\theta^\top z_s) + (1 - c_s)(1 - \sigma(\theta^\top z_s)) \right) . \tag{3}$$

Let $\Delta_{ij} > 0$ lower bound the margin of comparison, $|\sigma(z_{ij}^\top \theta_*) - 1/2| > \Delta_{ij}$ for all $i, j \in \mathcal{I}$ and define $\Delta_* = \min_{i \neq j} \Delta_{ij}/|i - j|$. Next, let $\mathbf{H}_t(\theta) := \sum_{s=1}^{t} \dot{\sigma}(z_s^\top \theta) z_s z_s^\top$ be the Hessian of the negative log-likelihood of observations at time $t$ under (2), given the parameter $\theta$, also known as *observed Fisher information*. We define $\tilde{\mathbf{H}}_t(\theta) := \frac{1}{t} \mathbf{H}_t(\theta)$. For a square matrix $V$, we define $\|x\|_V = \sqrt{x^\top V x}$. We make the following assumptions for our analysis:

**Assumption 1.** $\theta_*$ satisfies $\|\theta_*\|_2 \leq S$ for some $S > 0$.

**Assumption 2.** $\forall i \in \mathcal{I}$, we have $\|x_i\|_2 \leq Q$ for $Q > 0$.

**Assumption 3.** $\mathbf{H}_T(\theta_T)$ and $\mathbf{H}_T(\theta_*)$ have full rank and minimum eigenvalues larger than $\lambda_0 > 0$.

Assumption 1 implies that $\theta_*$ lies in some ball with radius $S$ and cannot have unbounded coefficients. Assumption 2 states that there exists an upper bound on the norm of the feature vectors. This assumption is trivially satisfied whenever we have a finite set of data points. Both assumptions 1 and 2 are standard in the bandit literature and only required for our analysis. Assumption 3 is naturally satisfied for sufficiently large $T$ by any sampling strategy with support on $d$ linearly independent vectors, or can be ensured by allowing for a burn-in phase of $d$ samples. Assumption 3 ensures the uniqueness of $\theta_t$.

We start by stating the following concentration result for the deviation of $\sigma(z_{ij}^\top \theta_T)$ from the true probability $\sigma(z_{ij}^\top \theta_*)$. Recall that, while queries $\mathcal{Q}_T$ (and therefore $\{z_t\}_{t=1}^T$) are fixed, the stochasticity in the feedback $c_t$ implies that $\theta_T$ and consequently $\tilde{\mathbf{H}}_T^{-1}(\theta_T)$ are random. The proof of Lemma 1 is found in Appendix C and builds on results for optimistic algorithms in logistic multi-armed bandits (Filippi et al., 2010; Faury et al., 2020).

**Lemma 1** (Concentration Lemma). *Define, for all pairs of items $i, j \in \mathcal{I}$, and any $\Delta > 0$,*

$$\alpha_{ij}(\Delta) := \exp\left( \frac{-\Delta^2 T}{8dC_1(\dot{\sigma}(z_{ij}^\top \theta_T)\|z_{ij}\|_{\tilde{\mathbf{H}}_T^{-1}(\theta_T)})^2} \right), \quad \beta_{ij}(\Delta) := \exp\left( \frac{-\Delta T}{dC_1\|z_{ij}\|_{\tilde{\mathbf{H}}_T^{-1}(\theta_T)}^2} \right).$$

*Then, if $\alpha := \alpha_{ij}(\Delta), \beta := \beta_{ij}(\Delta)$ and $\alpha, \beta \leq \frac{1}{4dT}$,*

$$P\left( |\sigma(z_{ij}^\top \theta_*) - \sigma(z_{ij}^\top \theta_T)| > \Delta \right) \leq 2dT (\alpha + \beta) .$$

*$C_1$ depends on $S, \lambda_0, Q$ from Assumptions 1–3 (see Appendix C for definition and proof).*

The concentration result in Lemma 1 is *verifiable* (given by observables) since the upper bound depends only on the maximum likelihood estimate $\theta_T$ at time $T$, not on $\theta_*$. We present a sharper, *unverifiable* bound in Appendix C which instead depends on $\theta_*$ but does not suffer from the explicit scaling with $d$ in the definitions of $\alpha$ and $\beta$. The bound in Lemma 1 can also be expressed in terms of $\mathbf{H}_T^{-1}(\theta_T)$ by using the equality $\|z_{ij}\|_{\mathbf{H}_T^{-1}(\theta_T)}^2 = \frac{1}{T}\|z_{ij}\|_{\tilde{\mathbf{H}}_T^{-1}(\theta_T)}^2$. As long as our sampling strategy ensures that the minimum eigenvalue of $\tilde{\mathbf{H}}_T(\theta_T)$ does not tend to zero, i.e., the strategy is *strongly consistent* (Chen et al., 1999), we have $\alpha_{ij}(\Delta_{ij}) \sim \exp[-\Delta_{ij}^2 T/(\dot{\sigma}(z_{ij}^\top \theta_T)^2\|z_{ij}\|_{\tilde{\mathbf{H}}_T^{-1}(\theta_T)}^2)]$ and

$\beta_{ij}(\Delta_{ij}) \sim \exp[-\Delta_{ij} T / \|z_{ij}\|^2_{\tilde{\mathbf{H}}_T^{-1}(\theta_T)}]$. Since $\Delta_{ij}^2 < \Delta_{ij} < 1/2$ by definition, we can view $\alpha$ as the *first-order* term and $\beta$ as the *second-order* term of our bound.

Lemma 1 formally captures the intuition that it should be easier to sort when annotations contain little noise, i.e., $\dot{\sigma}(z_{ij}^\top \theta_T)$ is small. Especially, we observe $\dot{\sigma}(z_{ij}^\top \theta_T) \approx 0$ for pairs where $\Delta_{ij}$ is sufficiently large, causing the first-order term to vanish, leaving us with the faster decaying second-order term $\beta$. Lemma 1 also tells us that the hardest pairs to guarantee a correct ordering for are the ones with both high *aleatoric* uncertainty under the MLE model, e.g., where annotators disagree or labels are noisy, captured by $\dot{\sigma}(z_{ij}^\top \theta_T)$, as well as high *epistemic* uncertainty captured by $\|z_{ij}\|_{\tilde{\mathbf{H}}_T^{-1}(\theta_T)}$.

A direct consequence of Lemma 1 is the following bound on the ordering error of $h_{\theta_T}$ over $\mathcal{I}$,

$$\mathbb{E}[R(\theta_T)] \leq \sum_{i \neq j} \frac{2 \min\{2dT \left(\alpha_{ij}(\Delta_{ij}) + \beta_{ij}(\Delta_{ij})\right), 1\}}{n(n-1)}.$$

The right-hand side in the above inequality can be bounded further by utilizing that $\Delta_{ij} \geq |i-j|\Delta_*$. Together with Markov's inequality, this yields the following bound on $P(R(\theta_T) \geq \epsilon)$.

**Theorem 1** (Upper bound on the ordering error).
*Let $\alpha_* := \max_{i \neq j} \alpha_{ij}(\Delta_*)$ and $\beta_* := \max_{i \neq j} \beta_{ij}(\Delta_*)$, with $\alpha, \beta$ from Lemma 1. Then, for $\alpha_*, \beta_* \leq \frac{1}{4dT}$ and any $\epsilon \in (0,1)$, the ordering error $R(\theta_T)$ satisfies*

$$P(R(\theta_T) \geq \epsilon) \leq \frac{4dT}{\epsilon n} \left( \left(\alpha_*^{-1} - 1\right)^{-1} + \left(\beta_*^{-1} - 1\right)^{-1} \right) \approx \frac{4dT}{\epsilon n} \left( \alpha_* + \beta_* \right),$$

*where $\alpha_*$ and $\beta_*$ decay exponentially with $T$.*

Theorem 1 suggests that the probability of $R(\theta_T) \geq \epsilon$ decays exponentially with a rate that depends on the quantities $\max_{i,j} \dot{\sigma}(z_{ij}^\top \theta_T) \|z_{ij}\|_{\tilde{\mathbf{H}}_T^{-1}(\theta_T)}$ and $\max_{i,j} \|z_{ij}\|^2_{\tilde{\mathbf{H}}_T^{-1}(\theta_T)}$. Both quantities are random variables that depend on the particular sampling strategy that yields $\mathbf{H}_T$. Focusing on the leading term, $\max_{i,j} \dot{\sigma}(z_{ij}^\top \theta_T) \|z_{ij}\|_{\tilde{\mathbf{H}}_T^{-1}(\theta_T)}$, Theorem 1 suggests that an active learner should gather data to minimize this quantity and obtain the smallest possible bound. The factor $\|z_{ij}\|^2_{\tilde{\mathbf{H}}_T^{-1}(\theta_T)}$ is the weighted norm of $z_{ij}$ w.r.t. the inverse of the observed Fisher information (cf. Kirsch and Gal (2022)). It controls the shape of the confidence ellipsoid around $\theta_T$ and the width of the confidence interval around $\theta_T^\top z_{ij}$. The leading term in Theorem 1 re-scales this quantity with aleatoric noise under the MLE estimate $\theta_T$. This suggests that higher epistemic (model) certainty is needed in directions with high aleatoric uncertainty—where item similarity increases noise in comparisons.

In Appendix C.3, we comment on i) generalizations to regularized preference models, ii) applications to generalized linear models with other link functions, iii) lower bounds on the ordering error, and iv) an algorithm-specific upper bound.

## 5 Greedy uncertainty reduction for ordering (GURO)

We present an active preference learning algorithm based on greedy minimization of the bound in Theorem 1, called GURO. We begin with fully contextual preference models of the form $\sigma(\theta^\top z_{ij})$ and return in Section 5.1 to parameterization variants to reduce the effects of model misspecification.

The main component of the bound in Theorem 1 to be controlled by an active learner is the term

$$\max_{i,j \in \mathcal{I}} \dot{\sigma}(z_{ij}^\top \theta_T) \|z_{ij}\|_{\mathbf{H}_T^{-1}(\theta_T)}, \tag{4}$$

which represents the highest uncertainty in the comparison of any items $i, j \in \mathcal{I}$ under the model $\theta_T$. A smaller value of (4) yields a smaller bound and a stronger guarantee. Recall that, for any $t = 1, ..., T$, $\theta_t$ is the MLE estimate of the ground-truth parameter $\theta_*$ with respect to the observed history $D_t$. Both factors in (4) are determined by the sampling strategy that yielded the item pairs $(i_t, j_t)$ in $D_T$ and, therefore, $\mathbf{H}_T$ and $\theta_T$ (the results of comparisons $c_{ij}$ are outside the control of the algorithm, but $z_{ij}$ are known).

Direct minimization of (4), for a subset $\mathcal{I}_D$, is not feasible without access to comparisons $c_{ij}$ and their likelihood under $\theta_T$. Instead, we adopt a greedy, alternating approach: In each round, a) a

**Algorithm 1** **G**reedy **U**ncertainty **R**eduction for **O**rdering (GURO), [BayesGURO]

---
**Require:** Training items $\mathcal{I}_D$, attributes $\mathbf{X} = \{x_i\}_{i \in \mathcal{I}_d}$
 1: Initialize $\theta_0$
 2: **for** $t = 1, ..., T$ **do**
 3:    Draw $(i_t, j_t)$ based on $\theta_t$ according to (5) [or (7)]
 4:    Observe $c_t$ from noisy comparison (annotator)
 5:    $D_t = D_{t-1} \cup \{i_t, j_t, c_t\}$
 6:    $\theta_t = \mathrm{MLE}(D_t)$ according to (3) [or $\theta_t = \mathrm{MAP}(D_t)$ as in (11) in the Appendix]
 7: **end for**
 8: Return $h_T$

---

single pair is sampled for comparison by maximizing (4) under the current model estimate, and b) $\theta_t$ is recomputed based on $D_t$. Specifically, at $t = 1, ..., T$, we sample,

$$i_t, j_t = \argmax_{i,j \in \mathcal{I}_D, i \neq j} \dot{\sigma}(z_{ij}^\top \theta_{t-1}) \|z_{ij}\|_{\mathbf{H}_{t-1}^{-1}(\theta_{t-1})} \,. \tag{5}$$

We refer to this sampling criterion as Greedy Uncertainty Reduction for Ordering (GURO), since it reduces the uncertainty of $\theta_t$ in the direction of $z_{ij}$. To see this, consider the change of $\mathbf{H}_t(\theta_t)$ after a single play of $i_t, j_t$. The Sherman-Morrison formula (Sherman and Morrison, 1950) yields,

$$\mathbf{H}_t^{-1}(\theta_{t-1}) = \mathbf{H}_{t-1}^{-1}(\theta_{t-1}) - \dot{\sigma}(z_t^\top \theta_{t-1}) \frac{\mathbf{H}_{t-1}^{-1}(\theta_{t-1}) z_t z_t^\top \mathbf{H}_{t-1}^{-1}(\theta_{t-1})}{1 + \dot{\sigma}(z_t^\top \theta_{t-1}) \|z_t\|_{\mathbf{H}_{t-1}^{-1}(\theta_{t-1})}^2} \,, \tag{6}$$

where $z_t := z_{i_t j_t}$. With $\xi$ as the second term in (6), it holds for all $i < j \in \mathcal{I}$, with $\mathbf{H}_{t-1} = \mathbf{H}_{t-1}(\theta_{t-1})$, that $\|z_{ij}\|_{\mathbf{H}_t^{-1}(\theta_{t-1})}^2 = \|z_{ij}\|_{\mathbf{H}_{t-1}^{-1}}^2 - \|z_{ij}\|_\xi^2 \leq \|z_{ij}\|_{\mathbf{H}_{t-1}^{-1}}^2$. The inequality is strict for the pair $i_t, j_t$ in (5). As $\theta_t$ converges to $\theta_*$, this pair becomes representative of the maximizer of (4) provided there is no major systematic discrepancy between $\mathcal{I}_D$ and $\mathcal{I}$.

Surprisingly, GURO can also be justified from a Bayesian analysis. Consider a Bayesian model of the parameter $\theta$ with $p(\theta)$ the prior belief and $p(\theta \mid D_t)$ the posterior after observing the preference feedback in $D_t$. A natural active learning strategy is to sample items $i_t, j_t$ for which the model preference is highly uncertain under the posterior distribution,

$$i_t, j_t = \argmax_{i,j \in \mathcal{I}_D, i < j} \hat{\mathbb{V}}_{\theta | D_{t-1}}[\sigma(\theta^\top z_{ij})] \,, \tag{7}$$

where $\hat{\mathbb{V}}_{\theta | D_{t-1}}[\sigma(\theta^\top z_{ij})]$ is a finite-sample estimate of the variance in predictions, computed by sampling from the posterior. In Appendix B.3, we show that the first-order Taylor expansion of the true variance is equal to the GURO criterion. Hence, we refer to sampling according to (7) as BayesGURO. Unlike GURO, BayesGURO can incorporate prior knowledge through $p(\theta)$ and benefits from controlled stochasticity through the empirical estimate $\hat{\mathbb{V}}$, which makes it appropriate for batched algorithms—a deterministic criterion would construct batches of a single item pair. Both GURO and BayesGURO are presented in Algorithm 1.

**Computational Complexity:** Running the algorithms requires $O(n^2)$ operations each iteration to evaluate the sampling criteria (Equation 5 or 7) on all possible pairs, a problem shared by many active preference learning algorithms (Qian et al., 2015; Canal et al., 2019; Houlsby et al., 2011). A way of mitigating this computational complexity is to, at each time step, sample a fixed number of comparisons and only evaluate on these, similar to the approach taken in Canal et al. (2019). When only looking at a sample of $m \ll n^2$ pairs, the complexity is reduced to $O(m)$. While making $m$ too small can hurt the sample complexity, we describe in Appendix E how we implemented this sub-sampling strategy to speed up computations in one of our experiments and observed no noticeable change in performance. Lastly, we want to highlight that in many realistic scenarios, the computational burden pales in comparison to the time it takes to query an annotator.

### 5.1 Preference models for in- and out-of-sample ordering

Our default preference model $h(i, j) = \mathbb{1}[f(i, j) > 0]$ is based on a *fully contextual* scoring function

$$f_\theta(x_i, x_j) = \theta^\top (x_i - x_j) \,, \tag{8}$$

fit with a logistic likelihood $\sigma(f(i,j)) \approx p(C_{ij} = 1)$. The model's strength is that the variance in its estimates grows with $d$, but not with $n = |\mathcal{I}|$, often resulting in quicker convergence than non-contextual methods for moderate dimension $d$ (see, e.g., Figure 2c). The fully contextual model also generalizes to unseen items as long as the attributes for $\mathcal{I}_D$ span attributes observed for $\mathcal{I}$.

The limitations of a fully contextual model are model misspecification (error due to the functional form), and noise (error due to $C$ not being fully determined by $X$). The former can be mitigated by applying the linear model to a representation function $\phi : \mathcal{X} \to \mathbb{R}^{d'}$, $f_\theta(x_i, x_j) = \theta^\top(\phi(x_i) - \phi(x_j))$. A good representation $\phi$, e.g., from a foundation model, can mitigate misspecification and admit different input modalities. As demonstrated in Figure 5 in the Appendix, even a representation pre-trained for a different task can perform much better than a random initialization.[2]

Noise due to insufficiencies in $X$ cannot be mitigated by a representation $\phi(x)$; If annotators consistently compare items based on features $U$ not included in $X$, no function $h(X_i, X_j)$ can perfectly order the items. However, for in-sample ordering of $\mathcal{I}_D$, adding per-item parameters $\zeta_i \in \mathbb{R}$ to the scoring function, one for each item $i \in \mathcal{I}_D$, can mitigate both misspecification and noise,

$$f_{\theta,\boldsymbol{\zeta}}(x_i, x_j) = \theta^\top(\phi(x_i) - \phi(x_j)) + (\zeta_i - \zeta_j). \tag{9}$$

We call this a *hybrid* model and apply it in "GURO Hybrid" and baselines in experiments. The term $\zeta_i - \zeta_j$ can correct the residual of the fully contextual model, which is small if a) the context captures the most relevant information about the ordering, and b) the functional form $\theta^\top \phi(x_i)$ is nearly well-specified. Using $\zeta_i - \zeta_j$ alone is sufficient in-sample, but has high variance (the dimension is $n$ instead of $d$) and poor generalization ($\zeta_i$ are unknown for items $i \notin \mathcal{I}_D$). In practice, we use L2 regularization to prevent the model from learning an arbitrary $\theta$ by using the full expressivity of $\zeta_i$ (see Appendix E for details). Empirically, our hybrid models exhibit the best of both worlds: When $\phi$ is poor, the model recovers and competes with non-contextual models (Figure 5); when $\phi$ is good, convergence matches fully contextual models (Figure 2).

## 6 Experiments

We evaluate GURO (Algorithm 1) and GURO Hybrid (see Section 5.1) in four image ordering tasks, one with logistic (synthetic) preference feedback, and three tasks based on real-world feedback from human annotators[3].We provide a synthetic experiment in Appendix E.2 that includes empirical estimates of the bound in Theorem 1. The experiments include five diverse baseline algorithms, described next. BALD (Houlsby et al., 2011) is *a priori* the strongest baseline since it is a contextual active learning algorithm, unlike the others. Its selection criterion greedily maximizes the decrease in posterior entropy, which amounts to reducing the epistemic uncertainty and includes a term to downplay the influence of aleatoric uncertainty. This is not always beneficial, as suggested by our analysis in Section 4, since learners may require several comparisons of high-uncertainty pairs to get the order right. CoLSTIM (Bengs et al., 2022) is a contextual bandit algorithm, developed for regret minimization and is not expected to perform well here. It is included to illustrate the mismatch between regret minimization and our setting.

TrueSkill (Herbrich et al., 2006; Graepel, 2012) is a non-contextual skill-rating system that models the score of each item as a Gaussian distribution, disregarding item attributes, and has been adopted in various works to score items based on subjective pairwise comparisons (Larkin et al., 2022; Naik et al., 2014; Sartori et al., 2015). We use the sampling rule from Hees et al. (2016), designed for ordering. Finally, we include Uniform sampling, and to illustrate the importance of accounting for aleatoric uncertainty, we use a version of GURO called NormMin that ignores the $\dot{\sigma}(z_{ij}^\top \theta_t)$ term and plays the pair maximizing $\|z_{ij}\|_{\mathbf{H}_t^{-1}(\theta_t)}$, i.e., it minimizes the *second-order* term in Lemma 1. NormMin corresponds to the selection criterion in the concurrent work Das et al. (2024), adapted to our problem of finding the correct ordering. We refer the reader to Appendix E.2 for a detailed comparison where NormMin performs significantly worse than Uniform on certain problem instances, and Appendix E for details regarding the implementation and the choice of hyperparameters for GURO, BayesGURO, and baselines.

---

[2]It is feasible to update representations during exploration (Xu et al., 2022; Singh and Chakraborty, 2021), but we do not consider that here.

[3]Our code is available at: https://github.com/Healthy-AI/GURO

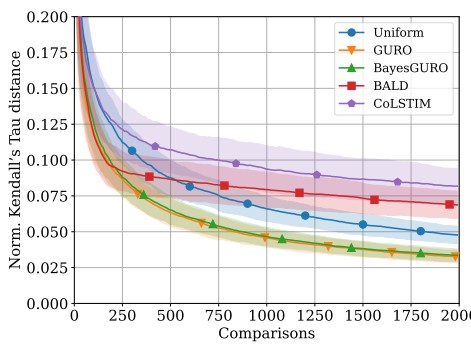 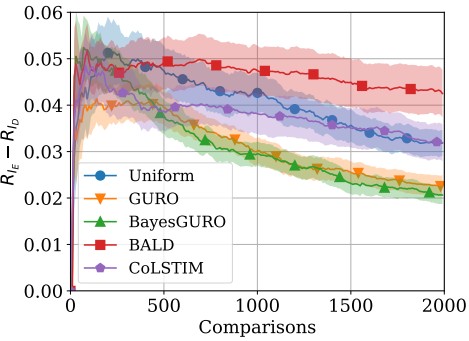

(a) Mean $R_{I_D}$ with 1-sigma error region.

(b) Mean generalization error (95% CI)

Figure 1: **X-RayAge**. Performance of active sampling strategies when comparisons are simulated using a logistic model according to (2). In-sample Kendall's Tau distance $R_{I_D}$ on 200 images (left) and generalization error $R_{I_E} - R_{I_D}$ for models trained on 150 images and evaluated on 150 images from a different distribution (right). All results are averaged over 100 different random seeds.

## 6.1 Ordering X-ray images under the logistic model

Our first task (X-RayAge) is to order X-ray images based on perceived age (Ieki et al., 2022) where the preference feedback follows a (well-specified) logistic model. We base this experiment on the data from the Kaggle competition "X-ray Age Prediction Challenge" (Felipe Kitamura, 2023) which contains more than $10\,000$ de-identified chest X-rays, along with the person's true age. Features were extracted using the 121-layer DenseNet in the TorchXrayVision package (Cohen et al., 2022) followed by PCA projection, resulting in 35 features. A ridge regression model, $\theta_*$, was fit to the true age ($R^2 \approx 0.67$). During active learning, feedback is drawn from $p(C_{ij} = 1) = \sigma\left(\theta_*^\top z_{i,j} \cdot \lambda\right)$, where $\lambda$ (set to 0.1) controls the noise level. We only include the fully contextual models here since they are well-specified by design, meaning $\mathcal{I}$ can be ordered using only contextual features.

In the first setting, we sub-sample 200 X-ray images uniformly at random from the full set. A ground-truth ordering of these elements is derived using the learned linear model. Figure 1a shows the ordering error over $2\,000$ iterations. GURO and BayesGURO perform similarly, both better than the baselines. BALD starts off converging about as fast as GURO, but plateaus, most likely as a result of actively avoiding comparisons with high aleatoric uncertainty—pairs where annotators disagree in their preferences. The poor performance of CoLSTIM highlights the discrepancy between regret minimization and recovering a complete ordering.

In the second setting, we evaluate how well the algorithms generalize to new items. First, we sample 300 X-ray images from the full dataset. Next, we split these into two sets, with one ($I_D$) containing the youngest $50\%$ and the other ($I_E$) the oldest $50\%$. The algorithms were then trained to order the list containing the younger subjects, but were simultaneously evaluated on how well they could sort the list containing the older subjects. The continuously measured difference in ordering error evaluated on $I_E$ and $I_D$ are presented in Figure 1b. While all algorithms are worse at ordering items in $I_E$, GURO and BayesGURO achieve the lowest average difference. Together with Figure 1a, this means that our proposed algorithms achieved the best in-sample and out-of-sample orderings. For completeness, the in-sample performance of algorithms in the generalization experiment in Figure 1b are included in Appendix E.2.

## 6.2 Ordering items with human preference data

Next, we evaluate our algorithm on three publicly available datasets to study the algorithms' performance when preference feedback comes from human annotators (see Table 1 for an overview, detailed information of datasets in Appendix E.1). The datasets are IMDB-WIKI-SbS (Pavlichenko and Ustalov, 2021), where annotators have stated which of two people appear older, ImageClarity (Zhang et al., 2016), where modified versions of the same image have been compared according to the level of distortion, as well as the extended WiscAds dataset (Carlson and Montgomery, 2017), where labels correspond to which political advertisement is perceived as more negative toward an

Table 1: Datasets with preference feedback from annotators. Pretrained models are ResNet34 (He et al., 2016), all-mpnet-base-v2 (Reimers and Gurevych, 2019), and FaceNet (Schroff et al., 2015).

| Dataset | $n$ | $d$ | #comparisons | Data type | Embedding Model |
|---|---|---|---|---|---|
| **ImageClarity** | 100 | 63 | 27 730 | Image | ResNet34 (Imagenet) |
| **WiscAds** | 935 | 162 | 9 528 | Text | all-mpnet-base-v2 |
| **IMDB-WIKI-SbS** | 6072 | 75 | 110 349 | Image | FaceNet (CASIA-Webface) |

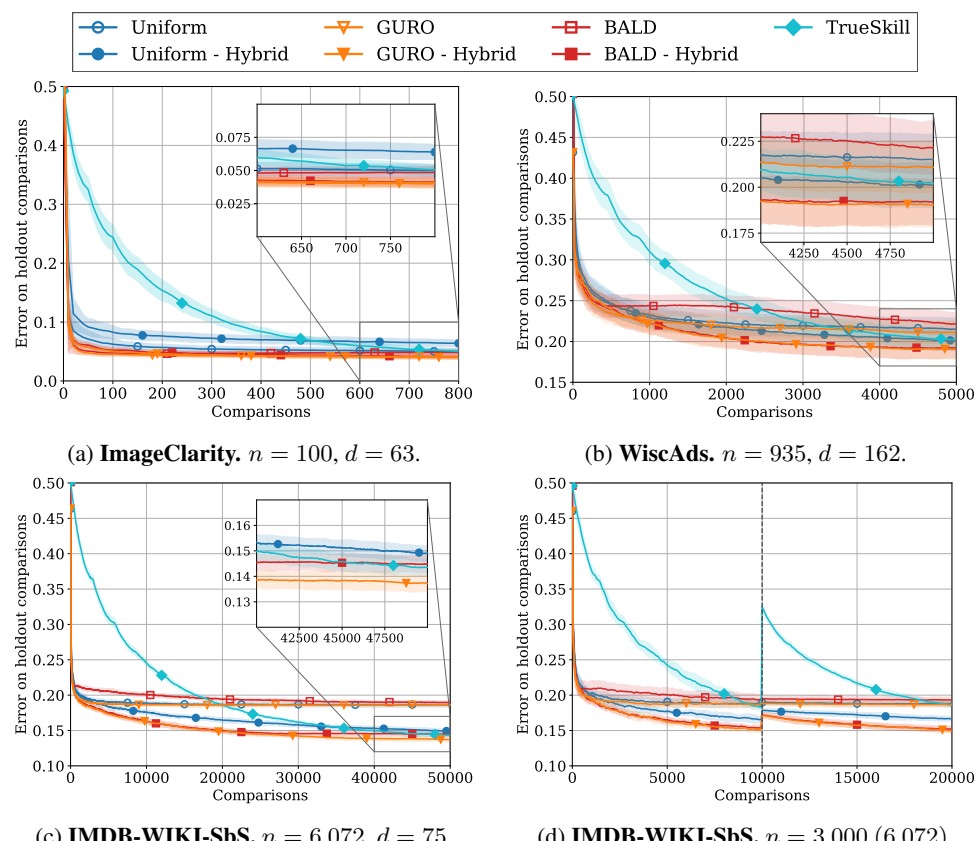

(a) **ImageClarity.** $n = 100$, $d = 63$.

(b) **WiscAds.** $n = 935$, $d = 162$.

(c) **IMDB-WIKI-SbS.** $n = 6\,072$, $d = 75$.

(d) **IMDB-WIKI-SbS.** $n = 3\,000\ (6\,072)$.

Figure 2: The empirical error $\hat{R}_{D'}(h)$ on a holdout comparison set $D'$ when comparisons are made by human annotators. The plots are averaged over 100 (a,b) or 10 (c,d) seeds, and the shaded area represents one standard deviation above and below the mean. For every seed, $10\%$ of comparisons were used for the holdout set. In (d) we initially order a list $\mathcal{I}_D$ of 3 000 images. After 10 000 comparisons the remaining 3 072 images, $\mathcal{I} \setminus \mathcal{I}_D$, are added.

opponent. In all datasets, pairs of items were sampled uniformly for annotation. For each experiment, we construct a feature vector $\phi(x_i) \in \mathbb{R}^d$ for all $n$ items using a pre-trained embedding model followed by PCA, applied to reduce computational complexity. We restrict algorithms to only query pairs for which an annotation exists and remove the annotation from the pool once queried. In cases where multiple annotations exist for the same pair, the feedback is chosen randomly among these.

The images in the ImageClarity dataset have been constructed to have an objective ground truth ordering but this is not the case for WiscAds or IMDB-WIKI-SbS. As the ground-truth ordering is generally unknown also in real-world applications, we evaluate methods by the error on a held-out set of comparisons $D'$, $\hat{R}_{D'}(h) = \frac{1}{|D'|} \sum_{(i,j,c) \in D'} \mathbb{1}[h(i,j) \neq c]$. This serves as an empirical analog of Kendall's Tau distance and a minimizer of $\hat{R}_{D'}(h)$ will minimize $R_{\mathcal{I}}(h)$ for sufficiently large $D'$, but will not converge toward 0 since there is inherent noise in annotations. This metric makes no assumptions on the ground truth ordering unlike the alternative approach of fitting an ordering to all available comparisons, see e.g., Maystre and Grossglauser (2017). In Appendix E.2,

we show results for the latter that highlight the limitations of estimating a "ground-truth" ordering, as well as the similar results when measuring the distance to the objective ground-truth ordering of the ImageClarity dataset. The longest trajectory (single seed) for any algorithm took less than 35hrs to complete on one core of an Intel Xeon Gold 6130 CPU and required at most 10 GB of memory.

In all experiments, we compare fully contextual (8) and hybrid (9) versions of GURO, BALD, and Uniform, as well as TrueSkill. The results of each experiment can be seen in Figure 2. Figure 2a shows that the ImageClarity dataset is the easiest to order using contextual (non-hybrid) features. This is expected, as features relevant to the level of distortion are low-level. In this case, the choice of adaptive strategy has a modest impact on the ordering error. Figures 2b and 2c highlight the differences between modeling strategies. The fully contextual algorithms initially improve rapidly, achieving a rough ordering of the items, before plateauing and not making any real improvements. This indicates that the features are informative enough to roughly order the list, but insufficient for retrieving a more granular ordering. The non-contextual TrueSkill converges at a much slower pace but keeps improving steadily throughout. Perhaps most interesting are the hybrid algorithms, which seemingly reap the benefits of both methods, improving as quickly as the contextual methods, but avoiding the plateau. In fact, in Figure 5 in the Appendix we show that the hybrid models perform comparably to TrueSkill even when features are completely uninformative.

The limitations of BALD are most noticeable in the fully contextual case, where it plateaus at a higher error compared to GURO and Uniform. This is however not as prominent when we use BALD in conjunction with our hybrid model, likely a result of the increased dimensionality of the model causing BALD Hybrid to attribute more of the observed errors to model uncertainty. While this initially causes the algorithm to avoid fewer comparisons that are subject to aleatoric uncertainty, the final iterations in Figure 2c suggest that BALD Hybrid can still run into this issue given enough samples. In all experiments, GURO and GURO Hybrid perform better than or similar to our baselines, never worse. Additionally, Figures 2b and 2c showcase how our hybrid model can increase performance when used with existing sampling strategies, such as BALD or Uniform.

The final experiment, visible in Figure 2d, is a few-shot scenario where after some time, additional images are added to the pool of items. IMDB-WIKI-SbS was used as it contained the highest number of both images and comparisons. The initial pool consists of 3 000 images sampled from the dataset. After 10 000 steps, the remaining 3 072 images were added to the pool. The results again emphasize the differences between our three types of models; the increase in error of the fully contextual model is very slight, likely a result of added samples being drawn from the same distribution. For TrueSkill, the error increases drastically as a result of the algorithm not having seen these items before and having no way of generalizing the results of previous comparisons to them. Lastly, the hybrid algorithms seem to be moderately affected. The error increases as the model has not yet tuned any of the added per-item parameters, but the extent is much smaller than for TrueSkill as the model can provide a rough ranking of the out-of-sample elements using the contextual features.

## 7 Conclusion

We have demonstrated the benefits of utilizing contextual features in active preference learning to efficiently order a list of items. Empirically, this leads to quicker convergence, compared to non-contextual methods, and allows algorithms to generalize out-of-sample. We derived an upper bound on the ordering error and used it to design an active sampling strategy that outperforms or matches baselines on realistic image and text ordering tasks. Both theoretical and empirical results highlight the benefit of accounting for noise in comparisons when learning from human annotators.

The optimality of our sampling strategy remains an open question. A future direction is to derive a lower bound on the ordering error, and prove an—ideally matching—algorithm-specific upper bound. However, constructing upper bounds for related fixed-budget tasks is an open problem (Qin, 2022). Moreover, motivated by the annotation setting, our focus has been on reducing sample complexity and we leave it to future work to explore potential linear approximations of the sampling criteria and other trade-offs between sample complexity and computational complexity. Further, our approach can potentially be improved by performing representation learning throughout the learning process. Finally, our experiments are constrained to a limited amount of already-collected (offline) human preference data, causing different algorithms to select disproportionately similar comparisons. Future work should evaluate the strategies in an online setting.

## Acknowledgements

FDJ and HB are supported by Swedish Research Council Grant 2022-04748. FDJ is also supported in part by the Wallenberg AI, Autonomous Systems and Software Program founded by the Knut and ALice Wallenberg Foundation. EC and DD are supported by Chalmers AI Research Centre (CHAIR). The computations were enabled by resources provided by the National Academic Infrastructure for Supercomputing in Sweden (NAISS), partially funded by the Swedish Research Council through grant agreement no. 2022-06725.

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

# A Notation

Table 2: Notation

| | |
|---|---|
| $\mathcal{I}$ | Collection of items $\mathcal{I} = \{1, ..., n\}$ |
| $n$ | Number of items |
| $d$ | Dimension of item attributes |
| $x_i \in \mathbb{R}^d$ | Context attributes for item $i \in \mathcal{I}$ |
| $z_{ij} \in \mathbb{R}^d$ | $z_{ij} = x_i - x_j$ for $i, j \in \mathcal{I}$ |
| $y_i$ | Score for item $i \in \mathcal{I}$ |
| $c_t \in \{0, 1\}$ | The outcome of the comparison at time $t$, 1 if $i_t$ was preferred to $j_t$ |
| $D_t$ | $D_t = ((i_1, j_1, c_1), ..., (i_t, j_t, c_t))$ |
| $\theta \in \mathbb{R}^d$ | Model parameter |
| $\theta_* \in \mathbb{R}^d$ | Model parameter of the environment |
| $\theta_t \in \mathbb{R}^d$ | Estimated parameter at time $t$ |
| $\sigma(.)$ | Sigmoid (logistic) function |
| $\dot{\sigma}(.)$ | derivative of $\sigma(.)$ |
| $\mathbf{H}_t(\theta)$ | Hessian of the negative log-likelihood $\mathbf{H}_t(\theta) := \sum_{s=1}^{t} \dot{\sigma}(z_s^\top \theta) z_s z_s^\top$ |
| $\tilde{\mathbf{H}}_t(\theta)$ | Hessian normalized by number of plays $\tilde{\mathbf{H}}_t(\theta) := \frac{1}{t} \mathbf{H}_t(\theta)$ |
| $\theta_{B,t} \in \mathbb{R}^d$ | The MAP estimate of $\theta$ at time $t$ |
| $\mathbf{H}_{B,t}$ | The Hessian in the Bayesian setting, adjusted by the prior covariance $\mathbf{H}_{B,0}^{-1}$ |
| $\|z_{ij}\|_{\mathbf{H}_t^{-1}(\theta)}$ | $\|z_{ij}\|_{\mathbf{H}_t^{-1}(\theta)} = \sqrt{z_{ij}^\top \mathbf{H}_t^{-1}(\theta) z_{ij}}$ |
| $S_t$ | $S_t := \sum_{s=1}^{t} \epsilon_s z_s$, where $\epsilon_s = c_s - \sigma\left(z_s^\top \theta_*\right)$ |
| $h$ | Comparison model (binary output) |
| $f$ | Comparison logit (typically linear), e.g., $f_\theta(i, j) = \theta^\top (x_i - x_j)$ |
| $S$ | An upper bound on the norm of $\theta_*$, that is $\|\theta_*\|_2 \leq S$ |

# B   Algorithms

## B.1   MLE estimator for logistic regression

The log-likelihood $L_t(\theta)$ of data $D_t = \{(i_s, j_s, c_s)\}_{s=1}^t$, with $z_s = x_{i_s} - x_{j_s}$, under a logistic regression model with parameters $\theta$ is defined by

$$L_t(\theta) = \sum_{s=1}^t \left( c_s \log \sigma(\theta^\top z_s) + (1 - c_s)(1 - \sigma(\theta^\top z_s)) \right) .$$

The maximum likelihood estimator (MLE) at time $t$ is the parameters

$$\theta_t = \arg\max_\theta L_t(\theta) . \tag{10}$$

The regularized estimator with ridge/$\ell_2$ penalty with parameter $\lambda$ is

$$\theta_t^R = \arg\min_\theta -L_t(\theta) + \lambda \|\theta\|_2^2 .$$

## B.2   Bayesian estimator for logistic regression

$\theta_{B,t}$ is the MAP estimate of $\theta$ at time $t$ according to the log likelihood

$$\theta_{B,t} = \arg\max_\theta \ln p(\theta \mid D_t), \tag{11}$$

where

$$\ln p(\theta \mid D_t) = -\frac{1}{2}(\theta - \theta_{B,0})^\top \mathbf{H}_{B,0}^{-1}(\theta - \theta_{B,0})$$
$$+ \sum_t c_t \ln(\sigma(z_{i_t,j_t}^\top \theta)) + (1 - c_t)\ln(1 - \sigma(z_{i_t,j_t}^\top \theta)) + const.$$

The hessian at time $t$ is defined as

$$\mathbf{H}_{B,t} = \mathbf{H}_{B,0} + \sum_{(i,j)\in D_t} \dot\sigma(z_{i,j}^\top \theta_{B,t}) z_{i,j} z_{i,j}^\top = \mathbf{H}_{B,0} + \mathbf{H}_t.$$

Moreover, if priors $\theta_{B,0} = \mathbf{0}$ and $\mathbf{H}_{B,0}^{-1} = I_d$ are used, the log likelihood boils down to:

$$\ln p(\theta \mid D_t) = -\frac{1}{2}\|\theta\|_2^2 + \sum_t c_t \ln(\sigma(z_{i_t,j_t}^\top \theta)) + (1 - c_t)\ln(1 - \sigma(z_{i_t,j_t}^\top \theta)) + const$$

which implies that the MAP estimate will be the same as the MLE estimate with ridge regularisation in the frequentist setting. Similarly, the Hessian becomes:

$$\mathbf{H}_{B,t} = \mathbf{H}_t + I_d$$

Sequential updates are also possible in the Bayesian setting by using your current estimates as the new priors. Note that this will give slightly different results, as the calculation of $\mathbf{H}_{B,t}$ depends on the current estimate of $\theta_{B,t}$.

## B.3   Stochastic Bayesian uncertainty reduction (BayesGURO)

We describe BayesGURO, a Bayesian sampling criterion, closely related to GURO. Consider a Bayesian model of the parameter $\theta$ with $p(\theta)$ the prior belief and $p(\theta \mid D_t)$ the posterior after observing the preference feedback in $D_t$. A natural strategy for learning more about the ordering of $\mathcal{I}$ is to sample items $i_t, j_t$ based on an estimate of the posterior variance of predictions for their comparison,

$$i_t, j_t = \arg\max_{i,j \in \mathcal{I}_D, i<j} \hat{\mathbb{V}}_{\theta|D_{t-1}}[\sigma(\theta^\top z_{ij})] . \tag{12}$$

Here, $\hat{\mathbb{V}}_{\theta|D_t}[\sigma(\theta^T z_{ij})]$ is an estimate of the variance of probabilities $\sigma(\theta^T z_{ij})$, computed from finite samples drawn from the posterior of $\theta$. Estimating the variance in this way both i) allows for tractable implementation, and ii) induces controlled stochasticity in the selection of item pairs. This can be useful in batched learning settings so that multiple pairs can be sampled within the same batch. A deterministic criterion would return the same item pair every time until $\theta$ is updated. We refer to the sampling criterion in (7) as BayesGURO.

For the logistic model considered in Section 4, using Laplace approximation with a Normal prior $\mathcal{N}(0, \mathbf{H}_{B,0}^{-1})$ on $\theta$, the Bayesian criterion in (7) is related to the GURO sampling criterion in (5) through the first-order Taylor expansion of the variance:

$$\mathbb{V}_{\theta|D_t}(\sigma(\theta^\top z_{ij})) \approx (\dot{\sigma}(\mathbb{E}_{\theta|D_t}[\theta^\top z_{ij}]))^2 \mathbb{V}_{\theta|D_t}[\theta^\top z_{ij}] = (\dot{\sigma}(\theta_{B,t}^\top z_{ij}) \|z_{ij}\|_{\mathbf{H}_{B,t}^{-1}(\theta_{B,t})})^2 \,,$$

where $\theta_{B,t}$ is the MAP estimate of $\theta$ at time $t$ and $\mathbf{H}_{B,t}$ is the Hessian adjusted by the prior covariance $\mathbf{H}_{B,0}^{-1}$ (further described in Appendix B.2). Thus, to a first-order approximation, for a large number of posterior samples, the GURO and BayesGURO active learning criteria are equivalent, save for the influence of the prior. In practice, we find that the Bayesian variant lends itself well to sequential updates of the posterior. The choice of prior $p(\theta)$, which could be useful under strong domain knowledge, and the stochasticity of using few posterior samples to approximate $\mathbb{V}$ make the two criteria distinct.

## B.4 Uniform sampling

The uniform sampling algorithm is given in Algorithm 2. The corresponding Bayesian version replaces line 5 with the MAP estimate.

---
**Algorithm 2** Uniform sampling algorithm
---
**Require:** Training items $\mathcal{I}_D$, attributes $\mathbf{X} = \{x_i\}_{i \in \mathcal{I}_d}$
1: **for** $t = 1, ..., T$ **do**
2:     Sample $(i_t, j_t)$ uniformly
3:     Observe $c_t$ from noisy comparison (annotator)
4:     $D_t = D_{t-1} \cup \{i_t, j_t, c_t)\}$
5:     Let $\theta_t = \text{MLE}(D_t)$
6: **end for**
7: Return $h_T$
---

## B.5 BALD

---
**Algorithm 3** BALD bandit
---
**Require:** Training items $\mathcal{I}_D$, attributes $\mathbf{X} = \{x_i\}_{i \in \mathcal{I}_d}$
1: Initialize $\theta_{B,0} = \mathbf{0}$, $\mathbf{H}_{B,0} = \lambda^{-1} I$
2: **for** $t = 1, ..., T$ **do**
3:     Draw $(i_t, j_t) = \arg\max_{i,j} H[y \mid z_{i,j}, D_{t-1}] - \mathbb{E}_{\theta \sim p(\theta|D_{t-1})}[H[y \mid z_{i,j}, \theta]]$
4:     Observe $c_t$ from noisy comparison (annotator)
5:     $D_t = D_{t-1} \cup \{i_t, j_t, c_t)\}$
6:     Let $\theta_t = \text{MAP}(D_t)$
7:     Update $\mathbf{H}_{B,t} \leftarrow \mathbf{H}_{B,0} + \sum_{(i,j) \in D_t} \dot{\sigma}(z_{i,j}^\top \theta_t) z_{i,j} z_{i,j}^\top$
8: **end for**
9: Return $h_T$
---

Where the posterior is calculated as in Appendix B.2 and $H[y \mid z_{i,j}, D_{t-1}] - \mathbb{E}_{\theta \sim p(\theta|D_{t-1})}[H[y \mid z_{i,j}, \theta]]$ is approximated as in Appendix B.5.1.

### B.5.1 Deriving the BALD sampling criterion

The BALD criteria formalized using our notation becomes

$$\arg\max_{i,j} H[y \mid z_{i,j}, D_t] - \mathbb{E}_{\theta \sim p(\theta|D_t)}[H[y \mid z_{i,j}, \theta]],$$

where $H$ represents Shannon's entropy

$$h(p) = -p \log_2(p) - (1 - p) \log_2(1 - p).$$

The first term of the equation becomes

$$H[y \mid z_{ij}, D_t] = h(\Pr(y \mid z_{i,j}, D_t)) = h\left(\int \Pr(y \mid z_{i,j}, \theta)\Pr(\theta \mid D_t)d\theta\right).$$

Here $\Pr(y \mid z_{i,j}, D_t)$ is the predictive distribution for our Bayesian logistic regression model. As covered in Bishop and Nasrabadi (2006, Chapter 4), this expectation cannot be evaluated analytically but can be approximated using the probit function $\Phi$;

$$\Pr(y \mid z_{ij}, D_t) \approx \Phi\left(\frac{\theta_t^\top z_{i,j}}{\sqrt{\lambda^{-2} + ||z_{ij}||^2_{\mathbf{H}_t^{-1}}}}\right) \approx \sigma\left(\frac{\theta_t^\top z_{i,j}}{\sqrt{1 + \frac{\pi ||z_{ij}||^2_{\mathbf{H}_t^{-1}(\theta_*)}}{8}}}\right).$$

Next, the term $\mathbb{E}_{\theta \sim p(\theta \mid D_t)}[H[y \mid z_{i,j}, \theta]]$ must be calculated. The true definition is

$$\mathbb{E}_{\theta \sim p(\theta \mid D_t)}[H[y \mid z_{i,j}, \theta]] = \int h(\sigma(\theta^\top z_{i,j}))\mathcal{N}(\theta \mid \theta_t, \mathbf{H}_t^{-1})d\theta.$$

To make this a one variable integral, let $X = \theta^\top z_{i,j}$ define a new random variable. Since $\theta \sim \mathcal{N}(\theta_t, \mathbf{H}_t^{-1})$, and $z_{i,j}$ is just a constant vector, we know that $X$ will follow a univariate normal distribution $X \sim \mathcal{N}(\theta_t^\top z_{i,j}, ||z_{ij}||^2_{\mathbf{H}_t^{-1}})$. This allows us to rewrite the integral as

$$\int h(\sigma(\theta^T \mathbf{z}))\mathcal{N}(\theta \mid \theta_t, \mathbf{H}_t^{-1})d\theta = \int h(\sigma(x))\mathcal{N}(\theta_t^\top z_{i,j}, ||z_{ij}||^2_{\mathbf{H}_t^{-1}})dx.$$

However, this integral has no closed form solution. Instead we perform the same strategy as in Houlsby et al. (2011) and do a Taylor expansion of $\ln h(\sigma(\theta^\top \mathbf{z}))$. The third-order Taylor expansion gives us

$$h(\sigma(x)) \approx \exp\left(-\frac{x^2}{8\ln 2}\right).$$

Inserting this, the term can be approximated as

$$\int h(\sigma(x))\mathcal{N}(x \mid \theta_t^\top z_{i,j}, ||z_{ij}||^2_{\mathbf{H}_t^{-1}})dx \approx \int \exp\left(-\frac{x^2}{8\ln 2}\right)\mathcal{N}(x \mid \theta_t^\top z_{i,j}, ||z_{ij}||^2_{\mathbf{H}_t^{-1}})dx$$

$$= \frac{C}{\sqrt{||z_{ij}||^2_{\mathbf{H}_t^{-1}} + C^2}}\exp\left(-\frac{(\theta_t^\top z_{i,j})^2}{2(||z_{ij}||^2_{\mathbf{H}_t^{-1}} + C^2)}\right),$$

where $C = \sqrt{4\ln 2}$. Finally, we arrive at an estimation of the objective function we wish to maximize:

$$H[y \mid z_{i,j}, D_t] - \mathbb{E}_{\theta \sim p(\theta \mid D_t)}[H[y \mid z_{i,j}, \theta]] \approx h\left(\sigma\left(\frac{\theta_t^\top z_{i,j}}{\sqrt{1 + \frac{\pi}{8}||z_{ij}||^2_{\mathbf{H}_t^{-1}}}}\right)\right)$$

$$- \frac{C}{\sqrt{||z_{ij}||^2_{\mathbf{H}_t^{-1}} + C^2}}\exp\left(-\frac{(\theta_t^\top z_{i,j})^2}{(||z_{ij}||^2_{\mathbf{H}_t^{-1}} + C^2)}\right)$$

# C   Proofs of Lemma 1 and Theorem 1

## C.1   Proof of Lemma 1

*Proof.* We now proceed to bound, under Assumptions 1–3 w.r.t. time $t$,

$$P\left(|\sigma(z_{ij}^\top \theta_t) - \sigma(z_{ij}^\top \theta_*)| > \Delta\right).$$

From the self-concordant property of logistic regression we have (Faury et al., 2020)

$$|\sigma(z_{ij}^\top \theta_t) - \sigma(z_{ij}^\top \theta_*)| \le \dot\sigma(z_{ij}\top\theta_t)|z_{ij}^\top(\theta_t - \theta_*)| + \frac{1}{4}|z_{ij}^\top(\theta_t - \theta_*)|^2. \tag{13}$$

We will prove a high probability bound on the event

$$\dot\sigma(z_{ij}^\top \theta_t)|z_{ij}^\top(\theta_t - \theta_*)| + \frac{1}{4}|z_{ij}^\top(\theta_t - \theta_*)|^2 \le \Delta. \tag{14}$$

Directly trying to bound the LHS in Equation 14 will result in a rather messy expression. Instead, we define the events

$$\mathcal{E}_1 := \left\{\dot\sigma(z_{ij}^\top \theta_t)|z_{ij}^\top(\theta_t - \theta_*)| \le \frac{\Delta}{2}\right\}$$

$$\mathcal{E}_2 := \left\{\frac{1}{4}|z_{ij}^\top(\theta_t - \theta_*)|^2 \le \frac{\Delta}{2}\right\}.$$

Clearly $\mathcal{E}_1 \bigcup \mathcal{E}_2$ implies the expression in Equation 14. Assume we have bounds on the complement of these events, $P(\mathcal{E}_1^c) \le \alpha$ and $P(\mathcal{E}_2^c) \le \beta$. Then

$$P\left(|\sigma(z_{ij}^\top \theta_t) - \sigma(z_{ij}^\top \theta_*)| > \Delta\right) \le \alpha + \beta + \alpha\beta$$

$$\le 2\alpha + 2\beta.$$

We now proceed to bound the probability of these complements separately.

**Step 1.   Relating $\theta_t$ to $\theta_*$:** The first challenge in our analysis to is relate $\theta_*$ and $\theta_t$. In contrast to linear regression, where we have a closed-form expression for $\theta_t$, there is no analytical solution for $\theta_t$ given a set of observation. However, we know that $\theta_t$ is the MLE, corresponding to

$$\theta_t = \arg\max_\theta L_t(\theta)$$

where

$$L_t(\theta) = \sum_{s=1}^t c_s \log\sigma\left(z_s^\top \theta\right) + (1 - c_s)\log\left(1 - \sigma\left(z_s^\top \theta\right)\right).$$

We have

$$\nabla_\theta L_t(\theta) = \sum_{s=1}^t c_s z_s - \underbrace{\sum_{s=1}^t \sigma\left(z_s^\top \theta\right) z_s}_{g_t(\theta)}$$

and hence $g_t(\theta_t) = \sum_{s=1}^t c_s z_s$.

A standard trick in logistic bandits (Filippi et al., 2010; Faury et al., 2020; Jun et al., 2021) is to relate $\theta_* - \theta_t$ to $g_t(\theta_*) - g_t(\theta_t)$. Especially, the following equality is due to the mean-value theorem (see Filippi et al. (2010))

$$g_t(\theta_*) - g_t(\theta_t) = \mathbf{H}_t(\theta')(\theta_* - \theta_t) \tag{15}$$

where $\theta'$ is some convex combination of $\theta_*, \theta_t$. Note that $\mathbf{H}_t(\theta')$ has full rank.

Using Equation 15 yields

$$\left|z_{ij}^\top(\theta_* - \theta_t)\right| = \left|z_{ij}^\top \mathbf{H}_t^{-1}(\theta')(g_t(\theta_*) - g_t(\theta_t))\right|$$

Furthermore, since $g_t(\theta_t) = \sum_{s=1}^t c_s z_s$, due to $\nabla_\theta L_t(\theta_t) = 0$, we have

$$g_t(\theta_t) - g_t(\theta_*) = \sum_{s=1}^t \underbrace{\left(c_s - \sigma\left(z_s^\top \theta_*\right)\right)}_{\epsilon_s} z_s$$

where $\epsilon_s$ is a sub-Gaussian random variable with mean 0 and variance $\nu_s^2 := \dot\sigma\left(z_s^\top \theta_*\right)$. We define

$$S_t := \sum_{s=1}^{t} \epsilon_s z_s.$$

We now have

$$\left| z_{ij}^\top (\theta_* - \theta_t) \right| = \left| z_{ij}^\top \mathbf{H}_t^{-1}(\theta') S_t \right|$$

and Lemma 10 in Faury et al. (2020) states that $\mathbf{H}_t^{-1}(\theta') \preccurlyeq (1+2S)\mathbf{H}_t^{-1}(\theta_*)$ where $||\theta_*||_2 \leq S$. Hence,

$$\left| z_{ij}^\top (\theta_* - \theta_t) \right| \leq (1+2S) \left| z_{ij}^\top \mathbf{H}_t^{-1}(\theta_*) S_t \right|$$

**Step 2. Tail bound for vector-valued martingales:**

We will now prove an upper bound on the probability that $\left| z_{ij}^\top \mathbf{H}_t^{-1}(\theta_*) S_t \right|$ deviates much from a certain threshold. This step is based on the proof of Lemma 1 in Filippi et al. (2010) which itself is based on a derivation of a concentration inequality in Rusmevichientong and Tsitsiklis (2010). The difference compared to Filippi et al. (2010) is that we work with the Hessian $\mathbf{H}_t(\theta_*)$ instead of the design matrix for linear regression $V_t = \sum_s x_s x_s^\top$. This require us to construct a slightly different martingale.

Let $A$ and $B$ are two random variables such that

$$\mathbb{E}\left[\exp\left\{\gamma A - \frac{\gamma^2}{2} B^2\right\}\right] \leq 1, \forall \gamma \in \mathbb{R} \tag{16}$$

then due to Corollary 2.2 in de la Peña et al. (2004) it holds that $\forall a \geq \sqrt{2}$ and $b > 0$

$$P\left(|A| \geq a\sqrt{(B^2 + b)\left(1 + \frac{1}{2}\log\left(\frac{B^2}{b} + 1\right)\right)}\right) \leq \exp\left\{\frac{-a^2}{2}\right\}. \tag{17}$$

Let $\eta \in \mathbb{R}^d$ and consider the process

$$M_t^\gamma(\theta_*, \eta) := \exp\left\{\gamma\eta^\top S_t - \gamma^2 ||\eta||_{\mathbf{H}_t(\theta_*)}^2\right\}. \tag{18}$$

We will now proceed to prove that $M_t^\gamma(\theta, \eta)$ is a non-negative super martingale satisfying Equation 16. Note that

$$\gamma\eta^\top S_t - \gamma^2 ||\eta||_{\mathbf{H}_t(\theta_*)}^2 = \sum_{s=1}^{t} \underbrace{\left(\gamma\eta^\top z_s \epsilon_s - \dot\sigma(\theta^\top z_s)\gamma^2 \left(\eta^\top z_s\right)^2\right)}_{F_s} = \sum_{s=1}^{t} F_s.$$

Further we use the fact that $\epsilon_s$ is sub-Gaussian with parameter $\nu_s$, .i.e,

$$\mathbb{E}\left[\exp\{\lambda\epsilon_s\}\right] \leq \exp\left\{\nu_s^2 \lambda^2\right\}, \forall \lambda > 0.$$

Let $D_{s-1}$ denote the observations up until time $s$, then

$$\mathbb{E}\left[\exp\{F_s\} \mid D_{s-1}\right] = \mathbb{E}\left[\exp\left\{\underbrace{\gamma\eta^\top z_s}_{\lambda} \epsilon_s\right\}\right] \exp\left\{-\underbrace{\dot\sigma(\theta_t^\top z_s)}_{\nu_s^2}\gamma^2 \left(\eta^\top z_s\right)^2\right\}$$

$$\leq \exp\left\{\nu_s^2 \lambda^2\right\} \exp\left\{-\nu_s^2 \lambda^2\right\} = 1.$$

This also implies

$$\mathbb{E}\left[M_t^\gamma(\theta_*, \eta) \mid D_{t-1}\right] \leq M_{t-1}^\gamma(\theta_*, \eta)$$

and $M_t^\gamma(\theta_*, \eta)$ is a super-martingale satisfying

$$\mathbb{E}\left[\exp\left\{\gamma\eta^\top S_t - \gamma^2 ||\eta||_{\mathbf{H}_t(\theta_*)}^2\right\}\right] \leq 1, \forall \gamma \geq 0$$

and we can apply the results of de la Peña et al. (2004).

We now follow the last step of the proof of Lemma 1 in Filippi et al. (2010). We let $a = \sqrt{2\log\frac{1}{\delta}}$ for some $\delta \in (0, 1/e)$ and let $b = \lambda_0 ||\eta||_2^2$. We have with probability at least $1 - \delta$

$$|\eta^\top S_t| \leq \sqrt{2\log\frac{1}{\delta}}\sqrt{||\eta||_{\mathbf{H}_t(\theta_*) + \lambda_0 ||\eta||_2^2}^2 \left(1 + \frac{1}{2}\log\left(1 + \frac{||\eta||_{\mathbf{H}_t(\theta_*)}^2}{\lambda_0 ||\eta||_2^2}\right)\right)}.$$

Rearanging and using the fact that $\lambda_0 ||\eta||_2^2 \le ||\eta||_{\mathbf{H}_t(\theta_*)}^2 \le t ||\eta||_2$ yields

$$|\eta^\top S_t| \le \rho(\lambda_0) ||\eta||_{\mathbf{H}_t(\theta_*)} \sqrt{2 \log \frac{t}{\delta}}. \tag{19}$$

where $\rho$ is defined as

$$\rho(\lambda_0) = \sqrt{3 + 2 \log \left( 1 + \frac{4Q^2}{\lambda_0} \right)}.$$

We take $M_t$ to be a matrix such that $M_t^2 = \mathbf{H}_t(\theta_*)$ and note that for any $\tau > 0$

$$P\left( ||S_t||_{\mathbf{H}_t^{-1}(\theta_*)}^2 \ge d\tau^2 \right) \le \sum_{i=1}^d P\left( \left| S_t^\top M_t^{-1} e_i \right| \ge \tau \right)$$

where $e_i$ is the i:th unit vector. Equation 19 with $\eta = M_t^{-1} e_i$ together with $||M_t^{-1} e_i||_{\mathbf{H}_t(\theta_*)} = 1$ yield that the following holds with with probability at least $1 - \delta$

$$||S_t||_{\mathbf{H}_t^{-1}(\theta_*)} \le \rho(\lambda_0) \sqrt{2d \log t} \sqrt{\log \frac{d}{\delta}}. \tag{20}$$

**Step 3. (Unverifiable) High-probability bounds on $\mathcal{E}_1$ and $\mathcal{E}_2$.**

We now have enough machinery to state high-probability bounds for our two events. These bounds will be *unverifiable* in the sense that the depend on the true parameter $\theta_*$ which is not known to us during runtime. We derive verifiable bounds in the next step of the proof.

Recall that $\mathbf{H}_t^{-1}(\theta_*)$ is symmetric. We apply Equation 19 with $\eta = \mathbf{H}_t^{-1}(\theta_*) z_{ij}$ and $\alpha > 0$ in place of $\delta$. First, we note that $||\mathbf{H}_t^{-1}(\theta_*) z_{ij}||_{\mathbf{H}_t(\theta_*)} = ||z_{ij}||_{\mathbf{H}_t^{-1}(\theta_*)}$ which implies with probability at least $1 - \alpha$

$$\left| z_{ij}^\top \mathbf{H}_t^{-1}(\theta_*) S_t \right| = \left| S_t^\top \mathbf{H}_t^{-1}(\theta_*) z_{ij} \right| \le \rho(\lambda_0) ||z_{ij}||_{\mathbf{H}_t^{-1}(\theta_*)} \sqrt{2 \log \frac{t}{\alpha}}. \tag{21}$$

We solve for smallest possible $\alpha \in (0, 1/e)$ such that

$$(1 + 2S) \rho(\lambda_0) \dot{\sigma}(z_{ij}^\top \theta_*) ||z_{ij}||_{\mathbf{H}_t^{-1}(\theta_*)} \sqrt{2 \log \frac{t}{\alpha}} \le \frac{\Delta}{2}$$

Rearranging yields

$$\alpha \le \exp \left\{ \frac{-\Delta^2}{8\rho^2(\lambda_0)(1+2S)^2 \left( \dot{\sigma}(z_{ij}^\top \theta_*) ||z_{ij}||_{\mathbf{H}_t^{-1}(\theta_*)} \right)^2} + \log t \right\}. \tag{22}$$

For $\mathcal{E}_2$ and the bound on its probability, $\beta > 0$ we have

$$\frac{1}{4} |z_{ij}^\top (\theta_t - \theta_*)|^2 \le \frac{1}{2} (1 + 2S)^2 ||z_{ij}||_{\mathbf{H}_t^{-1}(\theta_*)}^2 \rho^2(\lambda_0) \log \frac{t}{\beta} \le \frac{\Delta}{2}$$

and

$$\beta \le \exp \left\{ \frac{-\Delta}{\rho^2(\lambda_0)(1+2S)^2 \left( ||z_{ij}||_{\mathbf{H}_t^{-1}(\theta_*)} \right)^2} + \log t \right\}. \tag{23}$$

Note that both Equation 22 and Equation 23 are under the assumption that the RHS satisfy $< 1/e$ since this is required in order to apply the results of de la Peña et al. (2004). As we discuss in the main text, these quantities are approaching zero as $O(te^{-t})$, ignoring various constants, for reasonable sampling strategies and will satisfy this condition eventually.

**Step 4. (Verifiable) High-probability bounds on $\mathcal{E}_1$ and $\mathcal{E}_2$.**

The bounds in the previous step depend on the true parameter $\theta_*$ which we do not have access to in practice. We again use Lemma 10 of Faury et al. (2020) together with Cauchy-Schwartz

$$|z_{ij}^\top (\theta_* - \theta_t)| = \left| z_{ij}^\top \mathbf{H}_t^{-1/2}(\theta') \mathbf{H}_t^{1/2}(\theta') S_t \right|$$
$$\le (1 + 2S) ||z_{ij}||_{\mathbf{H}_t^{-1}(\theta_t)} ||S_t||_{\mathbf{H}_t^{-1}(\theta_*)}.$$

Connecting to Equation (13), we have

$$|\sigma(z_{ij}^\top \theta_t) - \sigma(z_{ij}^\top \theta_*)| \leq \dot{\sigma}(z_{ij}\top\theta_t)|z_{ij}^\top(\theta_t - \theta_*)| + \frac{1}{4}|z_{ij}^\top(\theta_t - \theta_*)|^2$$

$$\leq \dot{\sigma}(z_{ij}^\top\theta_t)(1 + 2S)||z_{ij}||_{\mathbf{H}_t^{-1}(\theta_t)}||S_t||_{\mathbf{H}_t^{-1}(\theta_*)}$$

$$+ \left(\frac{1}{4}(1+2S)||z_{ij}||_{\mathbf{H}_t^{-1}(\theta_t)}||S_t||_{\mathbf{H}_t^{-1}(\theta_*)}\right)^2.$$

Using Equation 20, and the fact that all terms are strictly greater than 0, we have with probability at least $1 - \alpha$

$$(1+2S)\dot{\sigma}(z_{ij}^\top\theta_t)||z_{ij}||_{\mathbf{H}_t^{-1}(\theta_t)}||S_t||_{\mathbf{H}_t^{-1}(\theta_*)} \leq (1+2S)\dot{\sigma}(z_{ij}^\top\theta_t)||z_{ij}||_{\mathbf{H}_t^{-1}(\theta_t)}\rho(\lambda_0)\sqrt{2d\log t}\sqrt{\log\frac{d}{\alpha}}. \tag{24}$$

We solve for smallest $\alpha \in (1/e)$ such that Equation 24 is smaller than $\Delta_{ij}/2$. This yields

$$\alpha \leq \exp\left\{\frac{-\Delta^2}{8d\rho^2(\lambda_0)(1+2S)^2\left(\dot{\sigma}(z_{ij}^\top\theta_t)||z_{ij}||_{\mathbf{H}_t^{-1}(\theta_t))}\right)^2} + \log dt\right\}.$$

Same steps for $\beta$ yields

$$\beta \leq \exp\left\{\frac{-\Delta}{d\rho^2(\lambda_0)(1+2S)^2\left(||z_{ij}||_{\mathbf{H}_t^{-1}(\theta_t))}\right)^2} + \log dt\right\}.$$

For brevity, define $C_1 = \rho^2(\lambda_0)(1+2S)^2$.

Using the definition of $\tilde{\mathbf{H}}_t$ yields the statement of Lemma 1. □

## C.2  Proof of Theorem 1

*Proof.* We let $i \succ j$ denote that $i$ is preferred to $j$. W.l.o.g assume $1 \succ 2 \succ ... \succ n$. The key observation is thatfor any $i$ and $j$ such that $i < j$ it holds that

$$\Delta_{i,j} > (j-i)\Delta_*.$$

If we get the wrong relation between $i, j$ then $\sigma(z_{ij}^\top\theta_*) - \sigma(z_{ij}^\top\theta_T) > (j-i)\Delta_*$. Lemma 1 implies

$$P(\sigma(z_{ij}^\top\theta_*) - \sigma(z_{ij}^\top\theta_T) > (j-i)\Delta) \leq dT(\exp\left\{\underbrace{\frac{-(j-i)\Delta^2}{8d\rho^2(\lambda_0)(1+2S)^2\left(\dot{\sigma}(z_{ij}^\top\theta_T)||z_{ij}||_{H_T^{-1}(\theta_T))}\right)^2}}_{\alpha_{ij}^{j-i}}\right\}$$

$$+ \exp\left\{\underbrace{\frac{-(j-i)\Delta}{d\rho(\lambda_0)(1+2S)^2\left(||z_{ij}||_{H_T^{-1}(\theta_T))}\right)^2}}_{\beta_{ij}^{j-i}}\right\}).$$

Let $R(\theta_T)$ be the ordering error of the $n$ items. Then, under a uniform distribution over items we have

$$\mathbb{E}[R(\theta_T)] \leq \frac{4dT}{n(n-1)}\left(\underbrace{\sum_{i=1}^{n-1}\sum_{j=i+1}^{n}\alpha_{ij}^{j-i}}_{A} + \underbrace{\sum_{i=1}^{n-1}\sum_{j=i+1}^{n}\beta_{ij}^{j-i}}_{B}\right) \tag{25}$$

$A$ and $B$ will be upper bounded using the same argument. We now upper bound sum $A$

Let $\alpha_* := \exp\left\{ \frac{-\Delta_*^2}{8dC_1 \max_{i,j} \dot\sigma(z_{ij}^\top \theta_T) \|z_{ij}\|^2_{\mathbf{H}_T^{-1}(\theta_*)}} \right\}$ then

$$A \leq \sum_{i=1}^{n-1} \sum_{j=i+1}^{n} \alpha_*^{j-i} \leq (n-1)\left( \sum_{j=0}^{n} \alpha_*^j - 1 \right)$$

$$\leq (n-1)\left( \frac{1}{1-\alpha_*} - 1 \right).$$

This follows from the definition of $\delta_{1,*}$ and properties of the geometric sum. It is easy to see that $\frac{1}{1-e^{-x}} - 1 = \frac{1}{e^x - 1}$. Hence,

$$\frac{4dT}{n(n-1)}A \leq \frac{4dT}{n}\left( \alpha_*^{-1} - 1 \right)^{-1}.$$

For $B$ we perform the same steps with $\beta_* := \exp\left\{ \frac{-\Delta_*}{dC_1 \max_{i,j} \|z_{ij}\|^2_{\mathbf{H}_T^{-1}(\theta_*)}} \right\}$ to get

$$\frac{4dT}{n(n-1)}A \leq \frac{4dT}{n}\left( \beta_*^{-1} - 1 \right)^{-1}.$$

Combing yields and

$$\mathbb{E}\left[R(\theta_T)\right] \leq \frac{4dT}{n}\left( \left(\alpha_*^{-1} - 1\right)^{-1} + \left(\beta_*^{-1} - 1\right)^{-1} \right)$$

By Markov's inequality we have

$$P(R(\theta_T) \geq \epsilon) \leq \frac{4dT}{\epsilon n}\left( \left(\alpha_*^{-1} - 1\right)^{-1} + \left(\beta_*^{-1} - 1\right)^{-1} \right). \tag{26}$$

$\square$

## C.3 Extensions of current theory

**Regularized estimators.** In our analysis in Section 4, we have assumed that $\theta_T$ is the maximum likelihood estimate and that $\mathbf{H}(\theta_T)$ has full rank. This can be relaxed by considering $\ell_2$ (Ridge) regularization where $\theta_{\lambda_0,T}$ is the optimum of the regularized log-likelihood with regularization $\lambda_0 \mathbf{I}$ and $\mathbf{H}_{\lambda_0}(\theta_{\lambda_0,T}) = \sum_{s=1}^{T} \dot\sigma(z_s^\top \theta_{\lambda_0,T}) z_s z_s^\top + \lambda_0 \mathbf{I}$. The same machinery used to prove Lemma 1 (Filippi et al., 2010; Faury et al., 2020) can be applied to this regularized version with small changes to the final bound.

**Generalized linear models.** It is also possible to derive similar results for generalized linear models with other link functions, $\mu(z_{ij}^\top \theta_*)$, by using the general inequality $\mathbf{H}(\theta) \geq \kappa^{-1}\mathbf{V}$ with $\mathbf{V} = \sum_{s=1}^{T} z_s z_s^\top$ and $\kappa \geq 1/\min_{z_{ij}} \dot\mu(z_{ij}^\top \theta_*)$. We conjecture that this will yield a scaling of $\sim \exp(-\Delta^2 T/\kappa)$ where, unfortunately, $\kappa$ might be very large. For a more thorough discussion on the dependence on $\kappa$ in generalized linear bandits, see Lattimore and Szepesvári (2020, Chapter 19).

**Lower and algorithm-specific upper bounds on the ordering error.** A worst-case lower bound on the ordering error can be constructed in the fixed-confidence setting, where the goal is to minimize the number of comparisons until a correct ordering is found with a given confidence, by following Garivier and Kaufmann (2016). This involves defining the set of *alternative* models $\text{Alt}(\theta_*)$ which differs from $\theta_*$ in their induced ordering of $\mathcal{I}$. The bound is then constructed by optimizing the frequency of comparisons of each pair of items so that such alternative models are distinguished as much as possible from the true parameter. We have left this result out of the paper as we find it uninformative in the regime when the number of comparisons is small, (see Simchowitz et al. (2017) for a discussion on the limitations of these asymptotic results in the standard bandit setting). Constructing a lower bound for our fixed-budget setting, of learning as good an ordering as possible with a fixed number of comparisons, is much more challenging. The fixed-confidence result yields *a* bound for the fixed-budget case (Garivier and Kaufmann, 2016), but constructing either a tight lower bound or a tight algorithm-specific upper bound is an open problem (Fang, 2022).

# D  Comparison with regret minimization

Bengs et al. (2022) considered a problem formulation where the goal is to learn a parameter $\theta$ which determines the utility $Y_{i,t}$ for a set of arms $i = 1, ..., n$ as a function of observed context vectors $x_{i,t}$ in a sequence of rounds $t = 1, ..., T$,

$$Y_{i,t} = \theta^\top X_{i,t} \ .$$

The probability that item $i$ is preferred over $j$ (denoted $i \succ j$) in round $t$ is decided through a comparison function $F$,

$$\Pr(i \succ j \mid X_{i,t}, X_{j,t}) = F(Y_{i,t} - Y_{j,t}) \ .$$

The goal in their setting is to, in each round, select two items $(i_t, j_t)$ so that their maximum (or average) utility is as close as possible to the utility of the best item. The expected regret in their average-utility setting is

$$\Re_{BSH} = \mathbb{E}[\sum_{t=1}^{T} 2Y_{i_t^*,t} - Y_{i_t,t} - Y_{j_t,t}] \ .$$

**Proposition 1** (Informal). *An algorithm which achieves minimal regret in the setting of Bengs et al. (2022) can perform arbitrarily poorly in our setting.*

*Proof.* The optimal choice of arm pair in the BSH setting is the optimal and next-optimal arm $(i_t^*, i_t')$ such that $i_t^* \succ i_t' \succ j$ for any other arms $j$. Assume that the ordering of all other arms $j$ is determined by a feature $X_{j,t}(k)$ but that $X_{i_t^*,t}(k) = X_{i_t',t}(k)$. Then, no knowledge will be gained about arms other than the top 2 choices under the BSH regret. As the number of arms grows larger, the error in our setting grows as well.  □

Saha (2021) study the same average-utility regret setting and give a lower bound under Gumbel noise. Saha and Krishnamurthy (2022) investigated where there is a computationally efficient algorithm that achieves the derived optimality guarantee.

# E Experiment details

For BayesGURO and BALD, the posterior $p(\theta \mid D_t)$ is estimated using the Laplace approximation as described in Bishop and Nasrabadi (2006, Chapter 4). With this approximation, the covariance matrix is the same as the inverse of the Hessian of the log-likelihood. For both methods, the priors $\theta_{B,0} = \mathbf{0}^d$ and $\mathbf{H}_{B,0}^{-1} = I_d$ were used, and sequential updates were performed every iteration. The sample criterion for BALD under a logistic model is given in Appendix B.5.1. For BayesGURO, 50 posterior samples were used to estimate $\hat{\mathbb{V}}_{\theta|D_t}[\sigma(\theta^T z_{ij})]$ for every $z_{ij}$. The hybrid algorithms follow the same structure with the added constraint that each per-item parameter $\zeta_i$ is independent of other parameters. This allows for efficient updates of $\mathbf{H}_{B,t}^{-1}$ by using sparsity in the covariance.

GURO, CoLSTIM, and Uniform use LogisticRegression from Scikit-learn (Pedregosa et al., 2011) with default Ridge regularization ($C = 1$) and the lbfgs optimizer. The former two updates $\theta_t$ every iteration using the full history, $D_t$ in all experiments except for IMDB-WIKI-SbS, where GURO updates $\theta_t$ every 25th iteration. This caused no noticeable change in performance as GURO still updates $\mathbf{H}_t^{-1}$ every iteration using the Sherman-Morrison formula. Note that when using the Sherman-Morrison formula in practice, you only get an estimate of $\mathbf{H}_t^{-1}(\theta_t)$ since previous versions have been calculated using older estimates of $\theta$. This method for approximating the inverse hessian is covered in Bishop and Nasrabadi (2006, Chapter 5) and when we compared it to calculating $\mathbf{H}_t^{-1}(\theta_t)$ from scratch every iteration we observed that the methods performed equally. The design matrix for CoLSTIM is updated as in Bengs et al. (2022): the confidence width $c_1$ was chosen to be $\sqrt{d \log(T)}$, and the perturbed values were generated using the standard Gumbel distribution.

To increase computational efficiency for the large IMDB-WIKI-SbS dataset, the hybrid algorithms did not evaluate all $\sim 100\,000$ comparisons at every time step. Instead, a subset of $5\,000$ comparisons was first sampled, and the highest-scoring pair in this set was chosen. This resulted in a large speed-up and no noticeable change in performance during evaluation.

## E.1 Datasets

**ImageClarity** Data available at `https://dbgroup.cs.tsinghua.edu.cn/ligl/crowdtopk`. This dataset contained differently distorted versions of the same image. To extract relevant features, we used a ResNet34 model (He et al., 2016) that had been pre-trained on Imagenet (Deng et al., 2009). After PCA projection feature dimensionality was reduced to $d = 63$. The dataset consisted of 100 images and $27\,730$ comparisons. Since the type of distortion is the same for all images, the dataset has a true ordering with regards to the strength of the distortion applied.

**WiscAdds** Data available at `https://dataverse.harvard.edu/dataset.xhtml?persistentId=doi:10.7910/DVN/0ZRGEE` (license: CC0 1.0). The WiscAdds dataset, containing 935 political texts, has been extended with $9\,528$ pairwise comparisons by Carlson and Montgomery (2017). In comparisons, annotators have stated which of two texts has a more negative tone toward a political opponent. To extract general features from the text, sentences were embedded using the pre-trained all-mpnet-base-v2 model from the Sentence-Transformers library (Reimers and Gurevych, 2019). After applying PCA to the sentence embeddings, each embedding had a dimensionality of $d = 162$.

**IMDB-WIKI-SbS** Data available at `https://github.com/Toloka/IMDB-WIKI-SbS` (license: CC BY). IMDB-WIKI-SbS consists of close-up images of actors of different ages. For each comparison, the label corresponds to which of two people appears older. The complete dataset consists of $9\,150$ images and $250\,249$ comparisons, but images that were grayscale or had a resolution lower than $160 \times 160$ were removed, resulting in $6\,072$ images and $110\,349$ comparisons. We extract features from each image using the Inception-ResNet implemented in FaceNet (Schroff et al., 2015) followed by PCA, resulting in $d = 75$ features per image.

## E.2 Additional figures

### X-RayAge

To highlight the importance of the first-order term in Lemma 1, we evaluated NormMin on the same X-ray ordering task as in Figure 1a. The results, shown in Figure 3a, indicate that not only does the algorithm perform worse than GURO, but is seemingly also outperformed by a uniform sampling strategy. Furthermore, for completeness, we include Figure 3b which shows the in-sample error, $R_{I_D}$, during the generalization experiment.

### Synthetic Example and Illustration of Upper Bound

In this setting, 100 synthetic data points were generated. Each data point consisted of 10 features, where the feature values were sampled according to a standard normal distribution. The true model, $\theta_*$, was generated by sampling each value uniformly between $-3$ and $3$. The pairwise comparison feedback was simulated the same

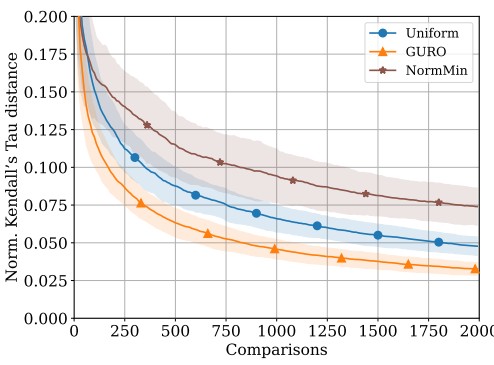
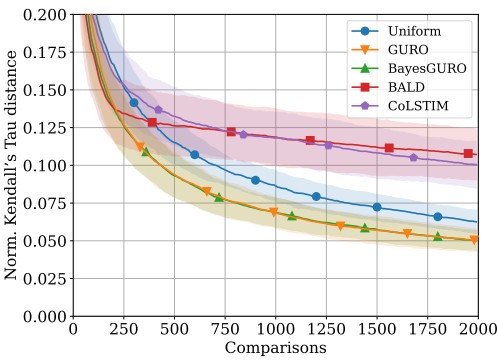

(a) **X-RayAge.** NormMin included in the experiment shown in Figure 1a.

(b) **X-RayAge.** The in-sample error $R_{I_D}$ for the generalization experiment performed in Figure 1b.

Figure 3: Additional figures from the **X-RayAge** experiment.

way as in Section 6.1, with $\lambda = 0.5$. The upper bound of the probability that $R(\theta_t) \geq 0.2$ was calculated every iteration according to Theorem 1. Each algorithm was run for 2000 comparisons, updating every 10th, the results of which can be seen in Figure 4. We observe in Figure 4b that our greedy algorithms are seemingly the fastest at minimizing the upper bound. The order of performance follows the same trend as in the experiments of Section 6.

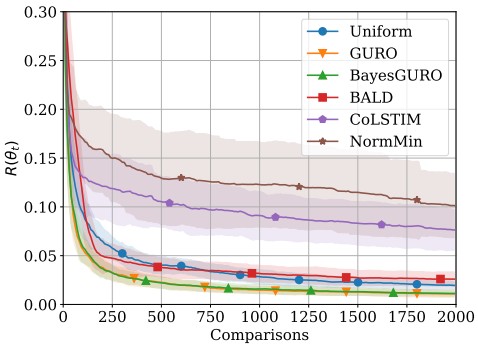
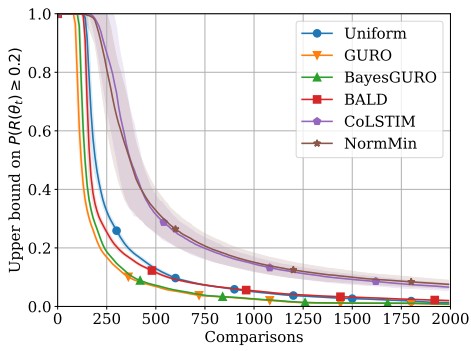

(a) The risk $R(\theta_t)$, defined as the normalized Kendall's tau distance between estimated and true orderings.

(b) The probability that the frequency of pairwise inversions is $\geq 20\%$ after every comparison, according to (1).

Figure 4: The loss (left) along with the upper bound (right) when ordering a list of size 100 in a synthetic environment. The results have been averaged over 50 seeds.

**Randomly initialized representation**

As discussed in Section 6.2, the performance of our contextual approach will depend on the quality of the representations. To underscore the practical usefulness of our algorithms, we have performed the same experiment as in Figure 2c, but this time the model used to extract image features was untrained (i.e., the weights were random). As to be expected, the results, shown in Figure 5, demonstrate that the fully contextual algorithms have no real way of ordering the items according to these uninformative features. However, GURO Hybrid performs similarly to TrueSkill, despite model misspecification. This is promising, since you may not know in advance how informative the extracted features will be for the target ordering task.

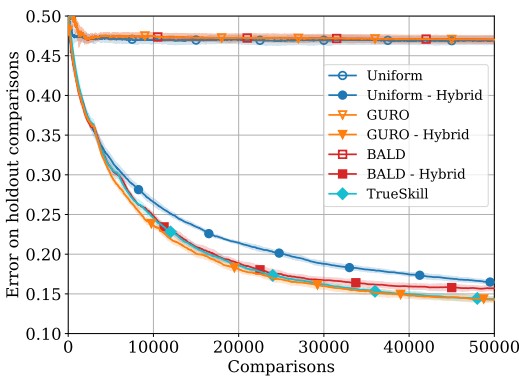

Figure 5: **IMDB-WIKI-SbS.** The same experiment as presented in Figure. 2c, but the model used for feature extraction is untrained.

**ImageClarity ground truth**

The ImageClarity dataset consists of multiple versions of the *same image*, with the *same distortion* applied to it to varying degrees. Due to this artificial construction, the pairwise comparisons should, given enough samples, reflect the magnitudes of the applied distortions. In Figure 6 we perform the same experiment as in Figure 2a, but instead of evaluating on a holdout comparison set, we measure the distance to the ground-truth ordering. The overall results are very similar, although we do see a slight increase in the performance of contextual algorithms compared to the non-contextual TrueSkill.

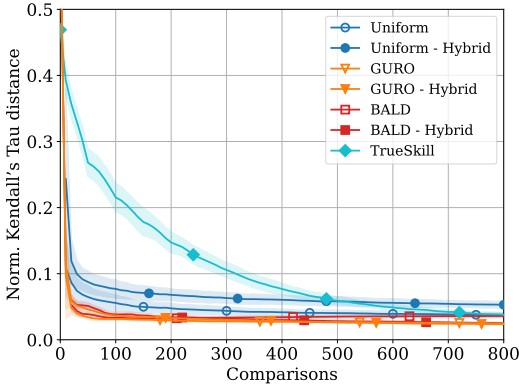

Figure 6: **ImageClarity.** Same experiment as in Figure 2a, but now measuring the distance to the ground-truth ordering. Averaged over 25 seeds along with the 1-sigma error region.

**Ground truth ordering using the Bradley-Terry model**

An alternate approach to evaluate ordering quality is to estimate a "ground-truth" ordering by applying the popular Bradley-Terry (BT) model (Bradley and Terry, 1952) to all available comparisons. We used the CrowdKit library (Ustalov et al., 2024) to find the MLE scores for each item and ordered the elements accordingly. In Figure 7 we run the same experiments as in Figure 2, but instead measure the distance to the constructed BT ordering. The overall trends remain, but for (b) and (c) there is a slight shift for the later iterations. More specifically we see non-contextual TrueSkill eventually overtaking the contextual algorithms.

The issue is that algorithms with orderings closer to the maximum likelihood estimate of the BT model will be favored. To exemplify this we use the ImageClarity dataset since it contains the largest number of comparisons relative to the number of items. We sample 1 000 comparisons and let this be the collection that is available to the algorithms. We further construct two target orderings, one from the BT estimate using the sampled subset of comparisons, and a second, more probable ordering, from the BT estimate using all 27 730 available comparisons. Figure 8 shows the distance between the GURO and TS algorithms and the different target orderings, where dashed lines indicate the distance to the ordering generated using the full list of comparisons. If we only look at the distance to the ordering produced using our subset of comparisons, TrueSkill seemingly outperforms GURO after about 350 comparisons. However, if we instead measure the distance to the more probable order-

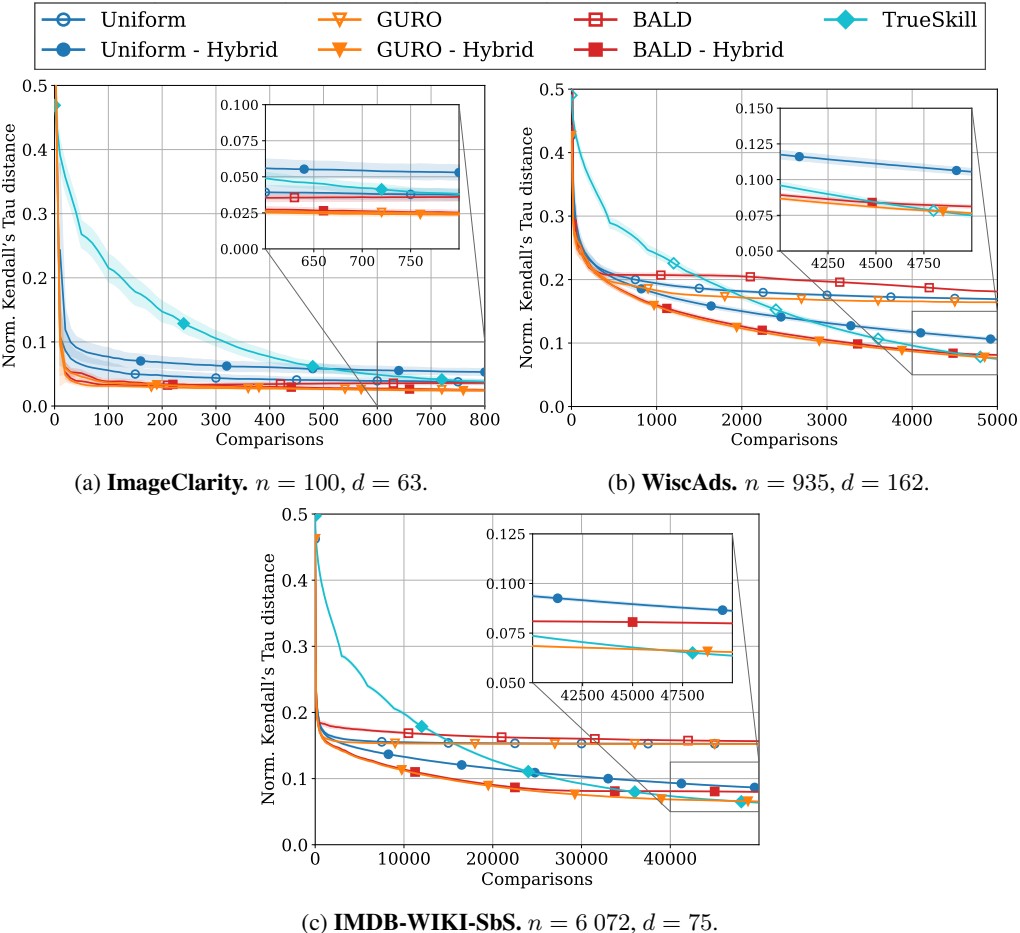

(a) **ImageClarity.** $n = 100, d = 63$.

(b) **WiscAds.** $n = 935, d = 162$.

(c) **IMDB-WIKI-SbS.** $n = 6\,072, d = 75$.

Figure 7: The same experiment as presented in Figure. 2c, but we instead measure the distance to a ground-truth estimated using all available comparisons.

ing, we see that GURO converges toward a lower distance. Note that these are the same orderings, evaluated against different targets. This is likely the effect we observe in Figure 7b and c, but not in Figure 7a as a result of the high amount of comparisons available to us.

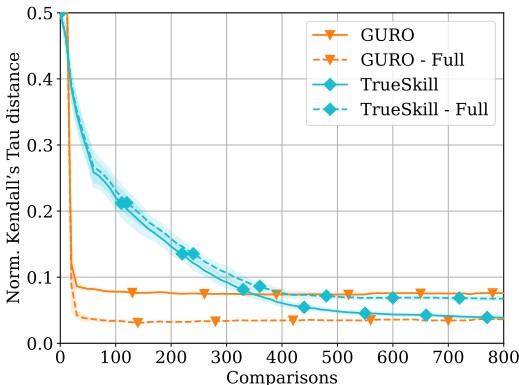

Figure 8: **ImageClarity.** The same experiment as presented in Figure 2a, but we instead measure the distance to target orderings that correspond to the maximum likelihood estimate of the BT model using different numbers of comparisons. The dashed lines show the distance to the BT estimate using all 27 730 comparisons.

