# OpenReview forum: "Active preference learning for ordering items in- and out-of-sample"
_NeurIPS.cc/2024/Conference — NeurIPS 2024 poster_

### Official Review · Reviewer_tuDy · 2024-07-02

**Soundness:** 3
**Presentation:** 3
**Contribution:** 2
**Rating:** 5
**Confidence:** 3

**Summary:**

This paper proposes an active learning algorithm for selecting pairs of items for comparison in order to learn a ranking of the items.
The ranking error is measured by the (normalized) number of swapped pairs compared to the true ordering (Kendall’s Tau distance). The algorithm chooses pairs of items based on an upper bound on this ranking error. Experiments are conducted on one synthetic and 3 real-world benchmarks.

**Strengths:**

* Active learning for pairwise comparisons is an interesting problem to study
* The approach is justified by theoretical analysis
* The empirical evaluation looks promising

**Weaknesses:**

* The paper uses strong assumptions on the pairwise comparisons. Specifically, the response is assumed to be: $P(C_{ij}=1) = \sigma(\theta_*\cdot(x_i-x_j))$, where $\sigma$ is the sigmoid function (eq 2). These assumptions are used for fitting MLE parameters. These assumptions are common in the literature.

* The optimization problem for choosing a pair (eq 5 and 7) seems hard. The cost is quadratic in the number of items since all pairs are considered. An approximation that depends linearly on the number of items would make the approach more practical.

**Questions:**

* “We restrict algorithms to only query pairs for which an annotation exists and remove the pair from the pool once queried.” – What is the number of available annotations for each dataset in figure 2 in the experiments? Only the number of items n and the dimension d are specified. Also, there is no notion of annotation noise here, since the labels are set, right?

* What is $\Delta_*$? It seems from line 159 that: $\Delta_*=\min_{i,j} \Delta_{ij}/|i-j|$, but it would be better to define it explicitly. Does it really depend on the index difference $|i-j|$? What is the reasoning behind this?

Minor:
* Line 74: “Ordering algorithms based only on preference feedback cannot solve this problem since observed comparisons are uninformative of new items.” This statement is not clear, as long as items are represented as attributes/features $x_i$ then comparisons may be informative for new items.

**Limitations:**

I suggest adding the assumption on response (eq 2) and the $O(n^2)$ complexity to the list of limitations.

---

> ### Author Rebuttal · Authors · 2024-08-06
>
> **R: The response is assumed to be: $P(C_{ij}=1) = \sigma(\theta_*(x_i-x_j))$.**
>
> While this assumption underlies our theoretical analysis and motivates the current version of GURO, we believe that we have taken many steps to highlight the practical usefulness of our algorithm and that the results are more generally applicable.
>
> * We describe in Appendix D.3 how one can derive similar results for other generalized linear models, meaning that one can extend these results beyond the logistic feedback setting.
>
> * We introduce a hybrid model that is aimed at overcoming model misspecification. In Figure 5 in Appendix F.2 we show that these hybrid models still perform comparably to non-contextual algorithms even when features are completely uninformative.
>
> * In Figure 2, we evaluate comparisons made by human annotators. While the algorithm uses a logistic model, the data does not adhere to this. Still, we observe good convergence properties. GURO outperforms TrueSkill, an algorithm that does not assume a logistic model.
>
> Finally, we thank reviewer tuDy for noting that these are common assumptions.
>
> **R: The optimization problem for choosing a pair (eq 5 and 7) seems hard.**
>
> The motivating example for this paper is the annotation of medical images. As annotations made by radiologists are expensive, our main goal has been to minimize sample complexity, or as stated in the introduction: ''... how can we learn the best ordering possible from a limited number of comparisons?''. We believe that the algorithm is already practically applicable as the bottleneck will in most cases be the time it takes the user to perform a comparison (e.g., about 6 seconds in the case of IMDB-WIKI-SbS). It is however true that for large collections, computational bottlenecks can be encountered if we evaluate all possible comparisons at every iteration, a problem shared by most active preference learning algorithms [1,2,3]. As we describe in Appendix F, in the cases where $n^2$ is large we can instead evaluate our sampling criterion on a uniform subsample of candidate comparisons with no noticeable impact on performance, similar to [2].
>
> Since the computational complexity of GURO was of interest to reviewer LY2M as well, we have written a discussion regarding this in the general rebuttal which we intend to include in a final version. This discussion also includes a potential alternative version of the sampling criterion that scales linearly with $n$.
>
> **R: There is no notion of annotation noise here, since the labels are set, right?**
>
>  The label for each **annotation** is set, but there is still noise in the **comparisons**. This is perhaps most clear for the ImageClarity dataset where each comparison has been annotated more than 5 times on average by different annotators. As these annotators can disagree with each other, we get different comparison outcomes depending on which annotation we sample. Furthermore, even in the case where we don't have multiple annotations for each comparison, we still observe noise since annotations can be inconsistent. Say the true order is $a \succ b \succ c$. We might observe the three annotations: $a \succ b$, $b \succ c$, and $c \succ a$. The final annotation is inconsistent with the best possible ordering, but we likely observe this in our data as human annotators disagree with each other and do not necessarily provide responses consistent with their previous comparisons. These inconsistencies are the reason for the error on the held-out comparisons not converging toward $0$ in Figure 2.
>
> **R: What is the number of available annotations for each dataset in figure 2?**
>
> The number of available comparisons for each dataset is provided in Table 1 (located above Figure 2) under "\#comparisons".
>
> * **ImageClarity** - 27 730
>
> * **WiscAds** - 9 528
>
> * **IMDB-WIKI-SbS** - 110 349
>
> **R: What is $\Delta_*$? Is it $\Delta_* = \min_{ij} \Delta_{ij}/|i-j|$?**
>
> You are correct, $\Delta_*$ should be defined as $\Delta_* = \min_{ij} \Delta_{ij}/|i-j|$ and we will add an explicit definition. Note, that we in the proof of Thm 1 assume, w.l.o.g, that the items are indexed such that $i > j$ implies $y_i > y_j$. This means that $|i-j|$ is the distance between the elements in the ordered list. Hence, we can lower bound $\Delta_{ij}$ by a constant times the difference in position in the ordered list. Substituting $\Delta_{ij}$ by $\Delta_* |i-j|$ in the lower bound on line 158 allow us to simplify that expression since all items in the sum now only depends on $\Delta_* |i-j|$ and we can apply results for geometric sums.
>
> **R: Clarification regarding ''algorithms based only on preference feedback'' on line 74.**
>
> As also pointed out by LY2M, this sentence can be improved. By "based only on preference feedback" we mean sorting algorithms that disregard attributes/features, regardless of whether they are present. This is the case for TrueSkill, and Figure 2d highlights its inability to generalize to new items. We will improve our formulation for the final version. For example, ''algorithms based only on preference feedback that ignore contextual information''.
>
> **R: I suggest adding the assumption on response (eq 2) and the $O(n^2)$ complexity to the list of limitations.**
>
> We thank the reviewer for this suggestion and believe that while we have addressed the former limitation in our first answer above, and will add a discussion on complexity to the final version, both topics warrant being mentioned as potential future directions to explore further.
>
> We thank tuDy for their suggestions for improving the text and have incorporated these into the updated manuscript.
>
> **References**
>
> * [1] Qian, et al. (2015). Learning user preferences by adaptive pairwise comparison.
>
> * [2] Canal, et al., (2019). Active embedding search via noisy paired comparisons.
>
> * [3] Houlsby, et al., (2011) Bayesian Active Learning for Classification and Preference Learning

---

> > ### Comment · Reviewer_tuDy · 2024-08-13
> >
> > Thank you for the clarifications.

---

### Official Review · Reviewer_6ajg · 2024-07-11

**Soundness:** 3
**Presentation:** 3
**Contribution:** 2
**Rating:** 5
**Confidence:** 3

**Summary:**

This paper considers the ranking problem based on pairwise comparisons. The goal is to get the best sampling strategy for the best ordering from a limited number of comparisons. Under a logistic model on the difference between scores, the authors provide the analysis for the upper bound on the ordering error, which provides insights on sample selections. Following the idea of minimizing the bound, this paper proposes the GURO algorithm for pair selections. The proposed method is evaluated in four image ordering tasks with either synthetic labels or real world labels.

**Strengths:**

The proposed algorithm is well motivated by the theoretical result on the ordering error. It has very strong theoretical guarantees on the performance. The theory presented in the paper looks good to me. And it helps the reader to understand the algorithm better with some justification from Bayesian analysis.
I find the paper very well written. The way the authors presented the results is very clear and easy to follow.

**Weaknesses:**

Although the theory presented in the paper looks good to me, I find it very similar to the result presented in the original Logistic bandit paper [1]. By treating the input space as the difference between features, the problem is simplified to a standard logistic bandit problem. And The result in Lemma 1 and some analysis before Theorem 1 are very similar to Lemma 2 and 3 in [1]. While I understand Theorem 1 is specifically for the ranking error, I think it is still straightforward to get Theorem 1 from existing lemmas.
And for the empirical study, I see the proposed approach actually does not always perform better than baselines, especially BALD. The good result is on synthetic data, there is no significant advance on real data. Therefore, I am not convinced with the claimed statement in the paper.

**Questions:**

Please provide more discussion on how difference the proposed method is comparing to logistic bandits.

**Limitations:**

Yes, the authors touch upon the limitation in the conclusion section.

---

> ### Author Rebuttal · Authors · 2024-08-06
>
> **R: Please provide more discussion on how difference the proposed method is comparing to logistic bandits.**
>
> As mentioned in our submission (l109, l138), our theory builds on the same techniques as previous papers on logistic bandits. However, we want to highlight that the problem considered here differs substantially from the standard bandit problem. Lemma 1 is a standard argument used in the literature (even before logistic bandits) which we include and tailor to our setting for completeness. Moreover, Theorem 1 is tailored toward our setting and justifies a selection criterion that is good for ranking but not for (regret minimization) bandits.
>
> Reviewer 6ajg did not specify which paper "[1]" is referring to but if it is Faury et al. (2020), the reviewer is correct that Lemma 1 is similar to their Lemma 3. An important distinction is that we provide a bound on the probability of error while they offer a high-probability upper bound on the prediction error. In principle, one can turn a bound on the prediction error to a bound on the probability of error. However, in the case of Lemma 3 in Faury et al. (2020), this would require solving a difficult expression involving squares of logarithmic terms. We are not sure this can be done in an analytical way that results in a parsable final bound. The reason that we can present a clean upper bound is due to the decomposition on l687 where we decompose the prediction error into first and second-order terms and then proceed to bound them independently. This is a new contribution in our work.
>
> **R: The proposed approach actually does not always perform better than baselines, especially BALD. [...] Therefore, I am not convinced with the claimed statement in the paper.**
>
> We assume that the reviewer is referring to the claim of superior sample efficiency. This claim refers to a comparison with previously published algorithms (Uniform, BALD, CoLSTIM, TrueSkill). In all cases in Figures 1-2, a method proposed in this work (GURO, GURO Hybrid or BALD Hybrid) performs better than these baselines, although the difference on ImageClarity is smaller. For the fully contextual versions, GURO consistently uses fewer samples than BALD to reach the same ordering quality, thus having superior sample efficiency. GURO Hybrid and BALD Hybrid are close but GURO Hybrid is never worse. Both are consistently better than TrueSkill. These are new results that were not known in previous research.
>
> In Section 6.2 we reason that the cause of the similar performance between BALD Hybrid and GURO Hybrid is the increased dimensionality due to item-specific terms leading to BALD attributing more of the errors to epistemic uncertainty. It is possible that we would eventually see a similar plateau for BALD Hybrid in 2b as in 2c, but due to us having a limited amount of pre-collected annotations, these experiments can't be extended much further without the overlap of comparisons selected by the algorithms becoming too large. We emphasize that the Hybrid model is a novel contribution presented in this work. It is a pragmatic solution to utilize contextual attributes even when they are not sufficient to produce a complete ordering.
>
> We agree that our claims regarding the empirical results can be clarified further in the abstract and the introduction, and will therefore take steps to:
>
> * Clarify what we mean by GURO having superior performance to active preference learning baselines.
>
> * Further highlight that BALD Hybrid is new to this work.
>
> Finally, beyond GURO and Hybrid variants, we believe that one of this paper's main contributions is the evaluation of existing algorithms when applied to recover a complete ordering. To the best of our knowledge, this is the first time BALD has been used explicitly to order a list. Our experiments, where we also apply our hybrid modification, offer insights into when this works well (BALD in Figure 2a, BALD Hybrid in 2b), and when this does not (BALD in all Figures except 2a, BALD Hybrid in 2c).
>
> We thank reviewer 6ajg for their critique and will use this input to clarify our claims and contributions for the final version of the paper.

---

### Official Review · Reviewer_nNXZ · 2024-07-13

**Soundness:** 2
**Presentation:** 3
**Contribution:** 2
**Rating:** 4
**Confidence:** 3

**Summary:**

Active preference learning is different from deriving a complete ordering from preferences. It focuses on “If we collect comparisons D_T, how good is the resulting model’s predicted ordering in the item set”.  The paper proposes a sampling method in the active learning scenario. Theoretical analysis is also provided.

**Strengths:**

--Preference learning is critical to many downstream tasks.

--The proposed method is somewhat novel.

--The theoretical analysis is provided.

--Experimental results are shown to verify its effectiveness.

**Weaknesses:**

--The assumption 1 and 2 are not so intuitive. It is better to illustrate an example.

--Baseline methods are weak. Though many related studies are mentioned in the related work section, performances of baselines are not shown in the experiments.

--The number of comparisons is  not reduced tremendously on the ImageClarity in Table1.

--Performances should be emphasized in terms of the prediction ordering quality.

**Questions:**

State-of-the-art baselines should be added for performance comparison.

**Limitations:**

Limitation should be added.

---

> ### Author Rebuttal · Authors · 2024-08-06
>
> **R: Baseline methods are weak. Though many related studies are mentioned in the related work section, performances of baselines are not shown in the experiments.**
>
> * We argue that the baselines we have included are state-of-the-art and come from diverse fields: Active Preference Learning, Logistic Bandits, Non-Contextual Sorting.
> Since there is limited work on recovering a complete ordering using contextual attributes, there are no well-established benchmarks to beat.
>
> * We have selected baselines to study three main questions: the effects of including contextual information, the impact of the sampling criterion and the difference between in-sample and out-of-sample ordering.  Most algorithms discussed in the related work section are **related** but are **not** designed to solve our problem, and would not give evidence for these questions.
>
> * Many algorithms would either perform poorly or do not work in the offline setting where we are limited to a subset of comparisons. [1] does not work on a subset of comparisons and assumes no noise. [2] account for noise by collecting $N$ annotations for *every* comparison pair, which is not resourceful and is not possible in most practical settings, including the tasks in our empirical study. Several methods in the related work section do not use contextual information, and TrueSkill (which is included in our study) is considered the state-of-the-art for this group. We welcome specific suggestions for stronger baselines to include.
>
> **R: Performances should be emphasized in terms of the prediction ordering quality.**
>
> We agree that obtaining a good ordering is the focus of this paper, but a ground-truth ordering does not exist for most datasets where comparisons have been made by human annotators since individual annotators often disagree and no annotator labels all pairs. In Figure 7 in Appendix F.2, we perform the same experiments as in Figure 2, but we instead measure the ordering error compared to an ordering estimated from all available comparisons. While this offers similar results, we discuss why this approach can be problematic and highlight this with an example below. In the case of ImageClarity, where the true ordering was available, we evaluate against this in Figure 6 in Appendix F.2. as well, and show that the results mirror our findings in Figure 2.
>
> **R: The assumption 1 and 2 are not so intuitive. It is better to illustrate an example.**
>
> We thank reviewer nNXZ for this suggestion and will include the motivation behind these assumptions in the updated manuscript.
>
> * Assumption 1 implies that the $\theta_*$ lies in some bounded ball and cannot have unbounded coefficients. We use this assumption on line 705 and we believe this assumption to be necessary since it wouldn't be possible to bound the maximum distance between some unknown $\theta'$ and the true $\theta_*$ otherwise. Note that this assumption is not limiting since the bound on $\theta_*$ is not used by our algorithms.
>
> * Assumption 2 states that there exists an upper bound on the norm of the feature vectors. This assumption is trivially satisfied whenever we have a finite set of data points.
>
> Both assumptions are standard in the literature and only required for the analysis.
>
> **R: The number of comparisons is not reduced tremendously on the ImageClarity in Table1.**
>
> This is true. We state in Section 6.2 that this is most likely a result of it being easy to order the images in ImageClarity based on their extracted features due to the low semantic level of image distortion. We still include this experiment to highlight that:
>
> * Hybrid models are sometimes not necessary.
>
> * There is still a clear difference between the contextual and non-contextual algorithms.
>
> * Uniform performs worse than active sampling criteria.
>
> We cannot expect any method to beat all other methods on every task, and we are happy to show examples where several methods work well.
>
> **R: Limitation should be added.**
>
> Is there a specific limitation you are referring to? In Section 7 we cover limitations, such as the lack of a lower bound, and future directions, such as applying representation learning and performing experiments in an online setting. We are happy to include other limitations in an expanded discussion for the final version.
>
> **References**
>
> * [1] Nir Ailon. Active learning ranking from pairwise preferences with almost optimal query complexity" Advances in Neural Information Processing Systems, 2011
>
> * [2] Kevin G Jamieson and Robert Nowak. "Active ranking using pairwise comparisons" Advances in neural information processing systems, 2011

---

### Official Review · Reviewer_LY2M · 2024-07-13

**Soundness:** 4
**Presentation:** 4
**Contribution:** 4
**Rating:** 8
**Confidence:** 3

**Summary:**

This paper considers the setting of learning an ordering between items according to some scoring rule. The assumption is that this ordering is determined by a contextual scoring rule, determined from the features of each item. Such contextual structure can aid in more rapidly learning an ordering, and generalizing to out of sample items. This ordering is learned from asking pairwise preferences to an oracle, reducing uncertainty about the total order. Since there are a large number of possible comparisons to be asked, active learning is deployed to only query labels for a subset of comparisons. A theoretical argument is made about the optimal balance between aleatoric and epistemic uncertainty to target in adaptive selecting queries, motivating the GURO adaptive sampling strategy and variants. The performance of GURO is demonstrated in several empirical experiments on simulated data and data collected (offline) from real humans.

**Strengths:**

I believe this paper is excellent - it is *clearly* written, has a great flow, and theoretical and empirical arguments are tied together nicely to motivate the problem setting, establish the problem fundamentals, convey mathematical intuition about uncertainty reduction, and justify active selection. There is also a robust set of experiments demonstrating GURO (and variants) in practice against baselines, along with explanatory discussion and implementation details. To my knowledge the analysis and algorithmic ideas here are *original*, and this is a *high quality* submission. Although not the centerpiece of this work, in an age where RLHF and efficiently learning from human preferences is paramount in training large models and ranking queries, work in active preference learning and the contributions made here are *significant*. There is also a robust and thorough appendix providing details and theoretical proofs (disclaimer: I have read the main paper in careful detail, but only skimmed the appendix). Overall, this is elegant, interesting, and impactful work (both theoretically and empirically).

**Weaknesses:**

I do not have any explicit weaknesses to list. Instead, I have a list of comments and questions below that I would like the authors to address. However I feel confident that these can be addressed during the rebuttal phase.

One comment is that there is no discussion (unless I missed it) about computational complexity stating and comparing the big-O computational complexity of GURO, its variants, and other selection methods. This would be an interesting and strengthening addition to this work.

**Questions:**

- I think the statement "Moreover, the set we want to order is often larger than the set of items observed during training—we may want to rank new X-rays in relation to previous ones. This cannot be solved using per-item scores alone." should be clarified. If one knew absolute scores for all items (regardless if they are observed in training), isn't it trivial to compute an ordering? Or did the authors mean that pairwise responses collected during training could not generalize outside of training, without example features to predict from? [Edit: this does seem to be clarified in Section 2, but should be made more clear in the introduction]
- I find the sentence "However, as we show in Section 4, learning this map to recover a complete ordering is distinct from the general preference learning setting, and existing algorithms lack theoretical justification for this application" to be vague and should be clarified. What exactly is the "general preference learning setting", versus learning a map to recover complete orderings? What does "this application" refer to? Which of these two settings are you concerned with here?
- I'm confused by line 159. Why can one lower bound $\Delta_{ij}$ by a factor depending on the index difference $\lvert i -j \rvert$? Aren't the indices arbitrary, and agnostic to the underlying geometry of the feature space? Does this mean that a simple index permutation would drastically change this $\Delta$ quantity?
- in line 166, should the dependence not be on $\theta_T$ rather than $\theta_*$?. See line 157 which uses $\theta_T$. line 169 also jumps back to $\theta_T$
- line 189 is missing an important point: by definition the entirety of $\mathcal{I}$ is unavailable, only $\mathcal{I}_D$ is available to select from. This should be commented on.
- I think line 198 is too vague: "As θt converges to θ∗, this pair becomes representative of the maximizer of (4) provided there is no major systematic discrepancy between ID and I." Can you comment more on what constitutes acceptable vs unacceptable discrepancies between I and I_D? This does bring up a problematic point: what if $I_D$ is not sufficiently representative of $I$? Line 215 starts to hint at this discussion but I think it needs to be elaborated on, ideally more formally
- in GURO Hybrid, how are these $\zeta_i$ parameters actually learned? Is it just a joint MLE on $\theta$ and $\zeta_i$? In this case, what prevents the model from learning an arbitrary $\theta$ (i.e., $\theta = 0$) and just using the full expressivity of $\zeta_i$? Is there some sort of regularization in practice?
- for completeness, can you include a figure in the Appendix showing the experiment in Fig 1b, but just plotting $R_{I_D}$ instead of the difference? It would be good to know how each algorithm does in an absolute sense on $I_D$
- figure 2 would benefit from a log y scale - it is very difficult to discern between methods

Minor:
- line 147 uses the notation $\widehat{H}$ instead of $\widetilde{H}$. I assume this is a typo
- missing left parentheses in line 5 of GURO algorithm
- the word BayesGURO in Algorithm 1 should also probably be colored green to show the association to (7) and (11)
- be careful with red and green as distinguishing colors for readers with color vision deficiency
- fix quotes on line 258

**Limitations:**

Yes

---

> ### Author Rebuttal · Authors · 2024-08-06
>
> **R: The definition of "general preference learning'' and its distinction from the current setting**
>
> We agree that this can be clarified further. This paper focuses on learning a map to recover a complete ordering, but we leverage active preference learning to achieve this.  By ''the general preference learning setting'' we refer to existing literature that adaptively samples pairwise comparisons, subsequently observing which of two items is preferred, to learn a comparison function $h(i, j)$. Much of this related work focuses on learning a parameter vector ($\theta$) [1,2,3]. Our approach is a special case of this, and while we are trying to learn $\theta$, we are only doing so as far as it helps us order our list of items. Existing work emphasizes getting good approximations of $\theta$, with [1] maximizing $\hat{\theta}^T \theta$, while [2,3] try to minimize $|\hat{\theta} - \theta|_2$. If the true vector $\theta$ is known, it is sufficient to order the list. However, reducing uncertainty in all directions will likely be wasteful; we do not care about the accuracy of $\theta$ in directions that do not help determine the ordering. As highlighted in Figures 1a and 2c in our paper, while BALD is constructed to efficiently learn parameters by sampling pairwise comparisons, it is ill-suited for full rank recovery.
>
> **R: In GURO Hybrid, how are these parameters actually learned? Is there some sort of regularization in practice?**
>
> Yes, the parameters are learned through joint MLE on $\theta$ and $\zeta_i$, with regularization, as you suggest. The regularization prohibits the algorithm from learning an arbitrary $\theta$. We cover experimental details, including regularization, in Appendix F. However, we agree that this information is essential for the hybrid model and will include this motivation in the main paper for the final version.
>
> **R: Line 198 is too vague: Can you comment more on what constitutes acceptable vs unacceptable discrepancies between $\mathcal{I}$ and $\mathcal{I_D}$?**
>
> To learn the ordering perfectly, the feature differences for pairs of items in $\mathcal{I_D}$ must span the space spanned by the feature differences for pairs in $\mathcal{I}$. This is true with high probability when $\mathcal{I_D}$ is a random subset of $\mathcal{I}$, the dimension $d$ is small relative to $|\mathcal{I_D}|$ and the variation in directions of $z_{ij}$ for $i,j \in \mathcal{I}$ is sufficient (i.e., each direction is sufficiently covered). An ``unacceptable'' case would be where one dimension of $z_{ij}$ has large variance for some pairs $i,j \in \mathcal{I}$ but is constant for pairs in $\mathcal{I_D}$. In this case, the component of $\theta$ in this dimension would not be learned consistently, and the model would generalize poorly to $\mathcal{I}$.
>
> **R: Clarification regarding sorting "using per-item scores alone" in the introduction**
>
> The assumption here is that you don't have any contextual features available (or that your sorting algorithm does not utilize them) and that per-item scores are not known from the beginning but have to be estimated by observing pairwise comparisons. In this scenario, when new items are added to the collection we have no way of estimating their underlying scores (apart from arbitrary initial values such as the average) before observing further comparisons where they are included. This is exemplified by TrueSkills performance in Figure 2d.
>
> **R: Line 159: Why can one lower bound $\Delta_{ij}$ by a factor depending on the index difference $|i-j|$?**
>
> This follows from the definition of $\Delta_* = \min_{i\neq j} \Delta_{ij} /(i-j)$. The indecies are actually not arbitrary but we assume, w.l.o.g, that the items are ordered such that $i > j$ implies $y_i > y_j$. This was stated in the appendix and we will move it to the main paper. The idea is that we can now lower bound $\Delta_{ij}$ by a constant, $\Delta_*$, times how close the two items are to each other in the ordered list, $|i-j|$. This allow us to simplify the lower bound stated on line 158 since each element in the sum now depends on $\Delta_* |i-j|$ and we can treat it as a geometric sum.
>
> **R: Discussion on the computational complexity of GURO and its variants**
>
> Thank you for suggesting this addition. We include a discussion on this topic in the general rebuttal since two reviewers raised the issue. We intend to add this discussion to the final version of the paper.
>
> **R: For completeness, can you include a figure in the Appendix showing the experiment in Fig 1b, but just plotting $R_{I_D}$ instead of the difference?**
>
> We agree with the reviewer that this would be good for completeness and will include this figure in the appendix for the final version. We have attached the produced figure to the general rebuttal.
>
> **R: in line 166, should the dependence not be on $\theta_T$ rather than $\theta_*$? See line 157 which uses $\theta_T$. line 169 also jumps back to $\theta_T$**
>
> Yes it should be $\theta_T$, thank you for pointing this out.
>
> **R: line 189 is missing an important point: by definition, the entirety of $\mathcal{I}$ is unavailable, only $\mathcal{I_D}$ is available to select from. This should be commented on.**
>
> Yes, this is correct, direct minimization of (4) would be impossible considering we only have access to a subset $\mathcal{I_D}$. We will note this in the final version of the paper.
>
> We thank LY2M for the thorough feedback and for believing that our contributions are of great value. The remaining comments (including minor ones) have been addressed in the updated manuscript.
>
> **References**
>
> * [1] Qian, et al. (2015). Learning user preferences by adaptive pairwise comparison. Proceedings of the VLDB Endowment.
>
> * [2] Canal, et al., (2019). Active embedding search via noisy paired comparisons. International Conference on Machine Learning.
>
> * [3] Massimino, and Davenport (2021). As you like it: Localization via paired comparisons. Journal of Machine Learning Research.

---

> > ### Comment · Reviewer_LY2M · 2024-08-12
> >
> > Thank you for your response. I am satisfied with these points and leave my review unchanged (at an 8). Also, please double check the definition of $\Delta_*$ in line 125. It is missing the quotient you defined above. It just says $\Delta_* = \min_{ij} \Delta_{ij}$

---

### Author Rebuttal · Authors · 2024-08-06

Dear reviewers and chairs, thank you for evaluating our work.

We are happy that a majority of reviewers found that the reasons to accept this paper outweigh the reasons to reject it. As strengths, the reviews pointed to the importance of the problem (3/4 reviews), the theoretical justification for the proposed algorithms (4/4), and the empirical evaluation (3/4).
We also thank the reviewers for asking clarification questions and suggesting improvements to strengthen the paper. We have addressed these in individual rebuttals to each review and are ready to incorporate the arguments in the final manuscript.
Moreover, we have attached a plot showing $R_{I_D}$ for the experiment in Figure 1b in the original paper, as requested by reviewer LY2M, for completeness.

Two reviewers asked about the computational complexity of the algorithms. We expand on this below but stress that sample complexity, not computational complexity, is the focus of this work.

Two main factors impact the computational complexity of GURO. The first is the selection of the next comparison. When sampling according to (5), the bottleneck is the calculation of $\lVert z_{ij} \rVert_{H_{t-1}^{-1}}$ for every possible comparison, which scales according to $O(d^2n^2)$ where $n=$ number of items, and $d=$ dimension of the features. As covered in Appendix F, to speed up computations for IMDB-WIKI-SbS, we only evaluate a subsample of all possible comparisons every iteration. This resulted in no noticeable change in performance and is similar to the approach taken in [1]. When only looking at a sample of $m \ll n^2$ combinations this complexity is reduced to $O(d^2m)$. Another interesting direction could be to first evaluate the model uncertainty of individual item scores $\lVert x_{i} \rVert_{H_{t-1}^{-1}}$, and then only evaluate (5) for the $k$ items with the highest uncertainty, giving a complexity of $O(d^2kn)$.

The second factor is the update of model parameters. This is done by solving a logistic regression where the computational complexity of each iteration is $O(ds)$ where $s =$ the number of samples collected. Each iteration also includes updating the inverse Hessian using the Sherman-Morrison formula with a complexity of $O(d^2)$. An interesting benefit of BayesGURO is that it allows for sequential updates of $\theta$, avoiding having to solve the logistic regression using all previously collected samples (which are instead embedded into the prior).

**References**

* [1] Canal, et al., (2019). Active embedding search via noisy paired comparisons. International Conference on Machine Learning.

---

### Decision · Program_Chairs · 2024-09-25

**Decision:**

Accept (poster)

**Comment:**

The paper tackles the problem of learning to rank items based on a contextual scoring rule and item feature vectors, from noisy dueling feedback, in the pure exploration setting. The paper is well written and the problem tackled is especially relevant due to the recent popularity of the RLHF setting. The claims of the paper are supported by both novel theoretical results as well as an empirical evaluation.

The main weaknesses of the paper tied to its direct practical relevance. In particular, the algorithm does not show a substantial performance gain over existing baselines and given its computational complexity is scaling as $O(N^2d^2)$, it is not clear the approach would scale to the size of the datasets encountered in the settings modelled in the paper. The authors propose an alternative lower-complexity algorithm in their rebuttal but it is not accompanied by the same guarantees as the approach in the paper.

Nonetheless, it is my and the reviewer's opinion that the submission presents sufficient merit to warrant acceptance.